# The Lock-in Hypothesis: Stagnation by Algorithm

**Tianyi Alex Qiu** [* 1]  **Zhonghao He** [* 2]  **Tejasveer Chugh** [3]  **Max Kleiman-Weiner** [4]

## Abstract

The training and deployment of large language models (LLMs) create a feedback loop with human users: models learn human beliefs from data, reinforce these beliefs with generated content, reabsorb the reinforced beliefs, and feed them back to users again and again. This dynamic resembles an echo chamber. We hypothesize that this feedback loop entrenches the existing values and beliefs of users, leading to a loss of diversity in human ideas and potentially the *lock-in* of false beliefs. We formalize this hypothesis and test it empirically with agent-based LLM simulations and real-world GPT usage data. Analysis reveals sudden but sustained drops in diversity after the release of new GPT iterations, consistent with the hypothesized human-AI feedback loop.

**Website**: `thelockinhypothesis.com`

## 1. Introduction

**Human-LLM Feedback Loops**  Frontier AI systems, such as large language models (LLMs) (Zhao et al., 2023), are increasingly influencing human beliefs and values (Fisher et al., 2024; Leib et al., 2021; Costello et al., 2024). This creates a self-reinforcing feedback loop: AI systems learn values from human data at pre- and post-training stages (Conneau & Lample, 2019; Bai et al., 2022; Santurkar et al., 2023), influence human opinions through their interactions, and then reabsorb those influenced beliefs, and so on. What equilibrium will this dynamic process reach?

Some argue that this dynamic creates a collective echo chamber (Glickman & Sharot, 2024; Sharma et al., 2024; Anderson et al., 2024). Experimental evidence, including from randomized controlled trials on AI usage, supports this idea (Glickman & Sharot, 2024; Sharma et al., 2024; Ren et al.,

2024; Peterson, 2024; Jakesch et al., 2023). However there remains a key gap in prior analyses. First, existing studies focus primarily on unidirectional AI influence on humans, neglecting the feedback loop arising from *mutual* influence. Second, a mechanistic explanation of the feedback loop has not yet been established. Finally, much of the evidence is from laboratory settings, rather than real-world usage patterns.

**The Lock-in Hypothesis**  We use *lock-in* to refer to a state where a set of ideas, values, or beliefs achieves a dominant and persistent position. During lock-in, the diversity of alternative beliefs diminishes until they are marginalized or vanish entirely (Gabriel & Ghazavi, 2021; Hendrycks & Mazeika, 2022; Qiu et al., 2024). In domains where objective truth is possible, a population may converge to false beliefs (e.g., geocentrism); in domains without objective truth, a population may converge towards harmful beliefs (e.g., slavery or racism). Group-level diversity loss (Liang et al., 2024; Padmakumar & He, 2023; Anderson et al., 2024) combined with the emergence of reinforcing feedback loops (Williams et al., 2024; Taori & Hashimoto, 2023; Hall et al., 2022; Shumailov et al., 2024) has already been documented across fields. For instance, bias can become amplified through human-AI interactions (Glickman & Sharot, 2024), while institutional and technological factors may further accelerate lock-in (Gabriel & Ghazavi, 2021).

While "lock-in" in the age of LLMs appears to be a possible outcome, there has been no known attempt to characterize its cause and effect on human-AI interaction. In this study, we make the following contributions:

- **Formalizing the Lock-in Hypothesis (§3):** We construct a formal Bayesian model for the lock-in hypothesis. We show that collective lock-in to a false belief is inevitable when both feedback loops and a moderate level of mutual trust are present in a community.

> **The Lock-in Hypothesis**: The feedback loop in human-AI interaction will eventually lead a population to converge on false beliefs. Such beliefs, once formed, are hard to change with opposing evidence, as feedback loops indiscriminately amplify confidence in existing individual and collective beliefs and humans develop trust in AI.

---

[*]Equal contribution  [1]Peking University  [2]University of Cambridge  [3]Independent  [4]University of Washington. Correspondence to: Tianyi Alex Qiu <qiutianyi.qty@gmail.com>, Zhonghao He <zh378@cam.ac.uk>.

*Proceedings of the $42^{nd}$ International Conference on Machine Learning*, Vancouver, Canada. PMLR 267, 2025. Copyright 2025 by the author(s).

- **Mechanistic Simulations (§4):** We conduct agent-based LLM simulations to demonstrate the pathway through which feedback loops lead to lock-in, and establish diversity loss as a metric for lock-in.

- **Hypothesis Testing on WildChat (§5):** We discover discontinuous yet sustained conceptual diversity loss in human messages of real-world LLM usage after the release dates of new GPT versions trained on new human data. It is the first real-world evidence supporting the existence of a human-LLM feedback loop that reinforces user beliefs. In some cases, we also observe increases in diversity over time. This may be due to the versatility of LLMs being better leveraged by users. The interaction between lock-in and exploration-oriented (and hence pro-diversity) forces in human-AI interaction remains an open problem for future research.

## 2. Related Work

**Echo Chambers in Recommender Systems** The closest analogy to the lock-in effects we describe in language models are the *echo chambers* created by recommender systems (RecSys). An echo chamber is created when the system entrenches and polarizes the opinions and preferences of users by repeatedly recommending content that only represents a single agreeable view (Cinelli et al., 2021). Such effects have been shown empirically in randomized controlled trials (Hosseinmardi et al., 2024; Piccardi et al., 2024; Luzsa, 2019; Gillani et al., 2018; Hobolt et al., 2024; Wolfowicz et al., 2023), observational studies (Bessi et al., 2016; Boutyline & Willer, 2017; Bright et al., 2020), and simulations (Mansoury et al., 2020; Kalimeris et al., 2021; Hazrati & Ricci, 2022; Carroll et al., 2022). However, opposing findings have also been reported (Dubois & Blank, 2018; Hosseinmardi et al., 2024; Brown et al., 2022), and systematic reviews have yet to reach a definite conclusion (Bruns, 2017; Terren & Borge-Bravo, 2021).

LLMs and their interaction with human users may have similar effects, but there are substantial differences. For example, RecSys recommendations are personalized (i.e., optimized for each user individually). In contrast, today's LLMs mostly optimize against all users collectively through both vanilla preference learning and pluralistic alignment methods (Ge et al., 2024; Jin et al., 2024). Personalized recommendations have been hypothesized as a culprit for polarization (Bessi et al., 2016; Hobolt et al., 2024), while, as will be demonstrated in later sections, the preference learning of LLMs may cause lock-in at the group level. Efforts have been made to personalize LLMs to each user's specific preferences (Tseng et al., 2024). However, doing so merely shrinks the size of the "echo chamber" from a collective chamber for all users to small chambers for each user individually and, as will be shown in §3, lock-in can

occur regardless of the number of agents involved.

**Influence of Language Models on Human Users** While LLMs are designed to assist human users, they often exert unintended influence over human opinions. Such influence has been established in co-writing interactions (Jakesch et al., 2023), LLM-powered search systems (Sharma et al., 2024), LLM-generated suggestions (Danry et al., 2024; Leib et al., 2021), and dialogues with LLM-powered chatbots (Salvi et al., 2024; Hackenburg et al., 2024; Potter et al., 2024; Fisher et al., 2024; Costello et al., 2024). Theories and simulations have been designed to explain the nature of this influence (Ren et al., 2024; Peterson, 2024). The mere fact that LLMs influence humans does not mean the influence is either harmful or irreversible. However, it does create the dynamic of *mutual influence* between LLMs and human users, and our focus on such dynamics and their consequences sets us apart from other works on LLM influence.

**Feedback Loops in Language Models** Feedback loops are not uncommon in the study of language models. *Model collapse*, the degradation of model performance when trained on model-generated data (Shumailov et al., 2024), results from model outputs being fed into its own training data. *In-context reward hacking*, where LLMs' pursuit of an objective at test-time creates negative side effects (Pan et al., 2024), results from models over-optimizing an objective in an iterative deployment loop. Here, however, we focus specifically on the feedback loop between LLM outputs and human preferences: LLMs iteratively learn from incoming human preference data (Dong et al., 2024; Chen et al., 2023), while influencing human preference with their output (Salvi et al., 2024; Hackenburg et al., 2024; Potter et al., 2024; Fisher et al., 2024). This gives rise to a dynamic where confirmatory communication reinforces beliefs and where human opinions are learned and indiscriminately repeated to humans in later interactions. As a result, there are downstream consequences: human subject experiments demonstrate loss of opinion diversity (Sharma et al., 2024; Peterson, 2024) and bias amplification (Ren et al., 2024; Glickman & Sharot, 2024).

However, (1) this evidence comes from artificially designed laboratory settings (e.g., binary classification tasks on images) unrepresentative of those in the wild, and (2) no known attempt has been made to give a mechanistic account that explains population-level lock-in from the low-level dynamics of iterated training. We aim to fill these gaps.

## 3. Formal Model

In this section, we construct an analytical model that formalizes the hypothesized lock-in phenomenon and its underlying cause. We formally define lock-in as the *irreversible*

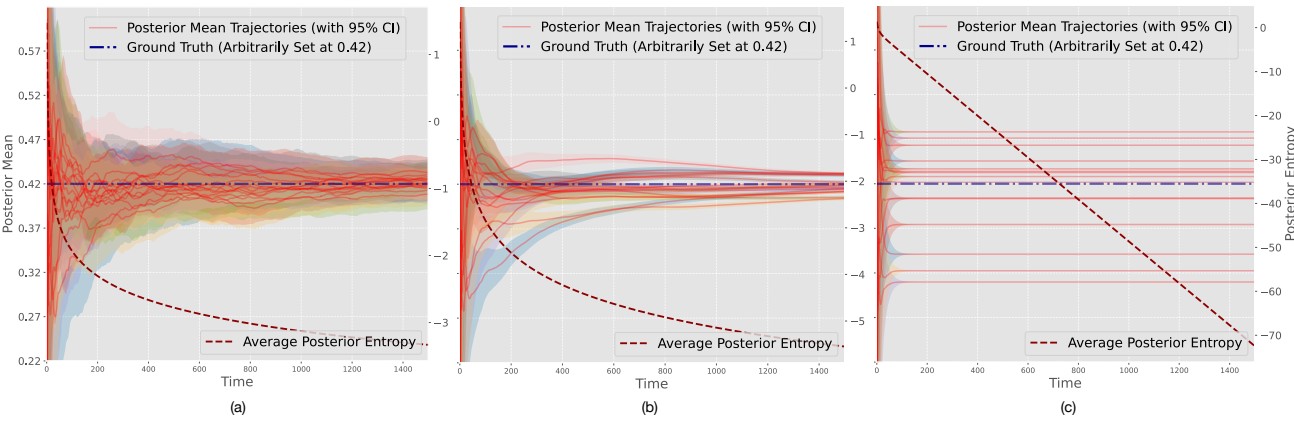

*Figure 1.* **Phase change of the Bayesian updating dynamics.** The critical threshold is $(N-1)\lambda_1\lambda_2 = 1$, where $N$ is the number of agents, and $\lambda_1, \lambda_2$ are degrees of mutual trust between human and AI agents. Each subfigure contains the trajectory of collective posterior belief over a certain target of estimation across 15 independent simulations, and the trajectory of the posterior entropy over time. **(a)** When $(N-1)\lambda_1\lambda_2 = 0.9$, collective beliefs of different runs converge towards the ground truth. **(b)** When $(N-1)\lambda_1\lambda_2 = 1.0$, convergence trends towards the ground truth remain but are accompanied by over-confidence. **(c)** When $(N-1)\lambda_1\lambda_2 = 1.1$, in every simulation, the collective posterior belief converge to a false value that's different from the ground truth.

*entrenchment of a belief*, characterize the conditions for lock-in, and develop *group-level diversity loss* as an observable metric of lock-in.

The setting is inspired by that of iterated learning, where individuals learn from others who learned similarly (Griffiths & Kalish, 2007; Kirby et al., 2014); and information cascades, where decisions based on others' actions rather than personal information lead to overconfident or false beliefs (Anderson & Holt, 1997; Zhou et al., 2021). In contrast to these models, we explicitly consider the topological structure of interactions, and, unlike iterated learning, we focus on the perils of mutual deference rather than the benefits of shared information. Our model is also related to belief propagation, a message-passing algorithm for computing marginals in graphical models (Su & Wu, 2015), but aims to model an epistemic process rather than carry out pobabilistic inference.

### 3.1. Basic Setup

Consider a group of $N$ agents, labeled $1, 2, \ldots, N$, tasked with estimating an unknown quantity $\mu \in \mathbb{R}$. At each time step $t$, agent $i$ measures $\mu$ with independent noise, i.e.

$$o_{i,t} \sim \mathcal{N}\big(\mu, \sigma_i^2\big),$$

where $\sigma_i^2$ is the variance of agent $i$'s measurement.

Based on these measurements, each agent $i$ maintains a *private* posterior about $\mu$, namely $\mathcal{N}\big(\hat{\mu}_{i,t}, p_{i,t}^{-1}\big)$, where $\hat{\mu}_{i,t} = \frac{1}{t}\sum_{i=1}^{t} o_{i,t}$ is the mean, and $p_{i,t} = t\sigma_i^{-2}$ is the precision (reciprocal of variance) of $i$'s posterior at time $t$.

For the sake of generality, we do not attempt to distinguish

between human agents and AI agents for now. Their distinction will be made later in the derivation.

Assume that an agent $i$ interacts with another agent $j$ and learns about its belief $\mathcal{N}\big(\hat{\nu}_{j,t}, q_{j,t}^{-1}\big)$. By aggregating its private belief and its secret access to $j$'s belief, $i$ arrives at its new *aggregate belief* $\mathcal{N}\big(\hat{\nu}_{i,t+1}, q_{i,t+1}^{-1}\big)$, where

$$\hat{\nu}_{i,t+1} = \frac{p_{i,t+1}\hat{\mu}_{i,t+1} + q_{j,t}\hat{\nu}_{j,t}}{p_{i,t+1} + q_{j,t}} \tag{1}$$

$$q_{i,t+1} = p_{i,t+1} + q_{j,t} \tag{2}$$

as a direct corollary of Bayes' theorem.

As is typical of both human-human and human-AI interactions, only $j$'s aggregate belief is accessible to $i$, while its private belief is kept to $j$ itself. Hence the recursive use of $\hat{\nu}_{j,t}, q_{j,t}$ when deriving $\hat{\nu}_{i,t+1}, q_{i,t+1}$.

### 3.2. The Trust Matrix and Transition Dynamics

Consider a 0-1 matrix $\mathbf{W} = (w_{i,j}) \in \{0,1\}^{N \times N}$, where $w_{i,j}$ denotes whether agent $i$ knows and trusts agent $j$'s posterior belief. Denote with $\hat{\boldsymbol{\mu}}_t, \hat{\boldsymbol{\nu}}_t, \boldsymbol{p}_t, \boldsymbol{q}_t \in \mathbb{R}^N$ the agent-wise parameter vectors at time $t$, and we have

$$\boldsymbol{p}_{t+1} = \boldsymbol{p}_t + \sigma^{-2}\mathbf{1} \tag{3}$$

$$\hat{\boldsymbol{\mu}}_{t+1} \odot \boldsymbol{p}_{t+1} = \hat{\boldsymbol{\mu}}_t \odot \boldsymbol{p}_t + \sigma^{-2}\boldsymbol{o}_t \tag{4}$$

$$\boldsymbol{q}_{t+1} = \boldsymbol{p}_{t+1} + \mathbf{W} \cdot \boldsymbol{q}_t \tag{5}$$

$$\hat{\boldsymbol{\nu}}_{t+1} \odot \boldsymbol{q}_{t+1} = \hat{\boldsymbol{\mu}}_{t+1} \odot \boldsymbol{p}_{t+1} + \mathbf{W}(\hat{\boldsymbol{\nu}}_t \odot \boldsymbol{q}_t) \tag{6}$$

where $\boldsymbol{o}_t \sim \mathcal{N}(\mu, \sigma^2\mathbf{I})$, and $\odot$ denotes pointwise product. (3) and (4) follow directly from definitions, while (5) and (6) are the multi-agent generalization of (2) and (1) respectively.

In the real world, people tend to discount opinions of others compared to their own. Extending the definition of $\mathbf{W}$, we may further allow its entries to take arbitrary non-negative real values. Given any $\mathbf{W} \in \mathbb{R}_{\geq 0}^{N \times N}$, we interpret $w_{i,j}$ as

- **When** $w_{i,j} = 0$: $i$ has no access to, or completely ignores, $j$'s aggregate belief.

- **When** $0 < w_{i,j} < 1$: $i$ sees $j$'s aggregate belief, but, believing that $j$'s measurements are noisier than it claims, discounts its precision $q_{j,t}$ by a factor of $w_{i,j}$.

- **When** $w_{i,j} = 1$: $i$ views $j$'s aggregate belief as equally trustworthy compared to its own.

- **When** $w_{i,j} > 1$: $i$ views $j$'s aggregate belief as more trustworthy than its own.

Transition dynamics (3)-(6) stay the same under this generalized *trust matrix* $\mathbf{W}$.

**Example 3.1** (Human-LLM Dynamics). *Consider a collection of one LLM advisor and $N - 1$ human users. We construct the trust matrix*

$$\mathbf{W} = \begin{pmatrix} 0 & \lambda_1 & \cdots & \lambda_1 \\ \lambda_2 & 0 & \cdots & 0 \\ \vdots & \vdots & \ddots & \vdots \\ \lambda_2 & 0 & \cdots & 0 \end{pmatrix},$$

*where the AI agent (labeled 1) trusts each human to the extent $\lambda_1 > 0$, representing its strength of preference learning; and the human agents (labeled 2 through $n$) each trust the AI agent to the extent $\lambda_2 > 0$. No communications exist between humans, resulting in zero entries.*

*Each human agent $i$ privately obtains observations $o_{i,t} \sim \mathcal{N}(\mu, \sigma^2)$. They each maintain both a private belief and an all-things-considered aggregate belief. The latter is secretly shown to the AI at each time step, who aggregates these beliefs and broadcasts the result. Each human agent updates their aggregate belief about $\mu$ based on both (1) their own private belief obtained from private measurements $o_{i,t}$ and (2) AI's aggregation of all agents' aggregate beliefs.*

Importantly, in Example 3.1, each human agent $i$ assumes that AI does not update on $i$'s belief (and thus AI's information serves as an independent source of information), while the AI does update its own belief based on $i$'s belief via preference learning, causing agent $i$ to double count its own beliefs. The result of such double-counting is then learned again by the AI, broadcasted to humans, double-counted again by others, etc. A **feedback loop** thus emerges.

Such a setting reflects the ignorance of how information flows in real-world interactions. LLMs are post-trained on human preference data, the latter erroneously assumed to be an independent source of truth uninfluenced by the LLM

itself (Dong et al., 2024; Carroll et al., 2024). Meanwhile, users perceive LLMs as objective "third parties," without necessarily discounting the ongoing preference learning process (Helberger et al., 2020; Glickman & Sharot, 2024).

### 3.3. Conditions for Lock-in

We start with a maximally general theorem, one that is agnostic towards human/AI distinctions. The proofs of both 3.2 and 3.3 can be found in Appendix C.

**Theorem 3.2** (Feedback Loops Induce Collective Lock-In). *Given any $\mathbf{W} \in \mathbb{R}_{\geq 0}^{N \times N}$, if the spectral radius $\rho(\mathbf{W}) > 1$, there exists $i \in \{1, \cdots, N\}$ such that*

$$\Pr\left[\lim_{t \to \infty} \hat{\mu}_{i,t} = \mu\right] = 0. \tag{7}$$

*Furthermore, when $\mathbf{W}$ is invertible and has spectral radius $\rho(\boldsymbol{W}) > 1$, (7) holds for all $i \in \{1, \cdots, N\}$.*

*When $\rho(\mathbf{W}) < 1$,*

$$\Pr\left[\lim_{t \to \infty} \hat{\mu}_{i,t} = \mu\right] = 1 \tag{8}$$

*for all $i \in \{1, \cdots, N\}$.*

In other words, feedback loops — where the circular flow of beliefs lead agents to unconsciously double-count evidence — can lead to false beliefs being permanently locked-in. Intuitively speaking, the condition $\rho(\boldsymbol{W}) > 1$ asks that the feedback loop be a *self-amplifying* one, instead of a *self-diminishing* one due to lack of trust between agents. See Theorem C.1 for an extension to a time-varying $\mathbf{W}$.

We now apply Theorem 3.2 to the specific human-LLM dynamics outlined in Example 3.1.

**Corollary 3.3** (Lock-in in Human-LLM Interaction). *Given any $N, \lambda_1 > 0, \lambda_2 > 0$, consider the following trust matrix representing human-LLM interaction dynamics.*

$$\mathbf{W} = \begin{pmatrix} 0 & \lambda_1 & \cdots & \lambda_1 \\ \lambda_2 & 0 & \cdots & 0 \\ \vdots & \vdots & \ddots & \vdots \\ \lambda_2 & 0 & \cdots & 0 \end{pmatrix}$$

*When $(N - 1)\lambda_1 \lambda_2 \leq 1$, for all $i \in \{1, \cdots, N\}$,*

$$\Pr\left[\lim_{t \to \infty} \hat{\mu}_{i,t} = \mu\right] = 1.$$

*When $(N - 1)\lambda_1 \lambda_2 > 1$, for all $i \in \{1, \cdots, N\}$,*

$$\Pr\left[\lim_{t \to \infty} \hat{\mu}_{i,t} = \mu\right] = 0.$$

Corollary 3.3 is validated with numerical simulations in Figure 1. A phase change is detected at $(N - 1)\lambda_1 \lambda_2 = 1$,

beyond which each simulation run converges exponentially to a false estimate of $\mu$. See Corollary C.3 for an extension to time-varying and heterogeneous trust parameters $\lambda_i$.

$(N - 1)\lambda_1\lambda_2 > 1$ is a relatively weak condition. When $N = 101$, it is only required that $\lambda_1, \lambda_2 > 0.1$ for Corollary 3.3 to apply — i.e., that humans and the AI discount each other's reported belief by a factor less than 10.[1] These results give a potential mechanistic account of lock-in: when feedback loops exist and are supported by mutual trust, collective lock-in to a confident false belief surely occurs.

It's worth noting that there are also forces that pull the human-AI system away from lock-in. These include sources of LLM capabilities that are independent of humans (Guo et al., 2025), human attempts at fact-checking (Guo et al., 2022), and more. It remains an open question how these forces interact with the human-AI feedback loop.

# 4. Simulations

## 4.1. Setup

In this section, we operationalize the lock-in hypothesis through a simulation, with the aim of learning how lock-in *might* happen and analyze what dynamic is responsible. This setup uses natural language to represent and communicate value-laden beliefs. As beliefs are expressed in natural language, this simulation aims to capture belief change mediated by LLMs. The simulation uses 100 agents, each simulated by GPT4.1-Nano. We instruct agents to hold initial beliefs on a given topic, consult a knowledge authority (i.e., the LLM), and then update their own beliefs. Each agent simulates a person learning from a centralized LLM authority. The authority is also simulated by GPT4.1-Nano, who is perceived to have expertise in the given topic. But in actuality, the authority acquires its beliefs from the group of agents.

Each agent maintains a belief statement in natural language over an *r/ChangeMyView* question. Example: "*I'm certain that* Citizens United was the worst thing to happen to the American political landscape." The authority sees and aggregates all agents' belief statements in-context, and broadcasts an aggregated belief statement in natural language. Agents then update beliefs in-context according to a pre-assigned trust in the authority (e.g., "high trust"). All four topics on which agents maintain beliefs are real posts from *r/ChangeMyView*: (1) "Discourse has become stupider, and as a result people are getting stupider, Since Trump was first elected in 2016."; (2) "Population decline is a great thing for future young generations."; (3) "Citizens United was the worst

---

[1] It doesn't mean the condition automatically holds for very large $N$, as people tend to downscale trust as the group size increases. A poll of 5 million people may exert less influence on a reader's opinion than a private discussion with 5 friends.

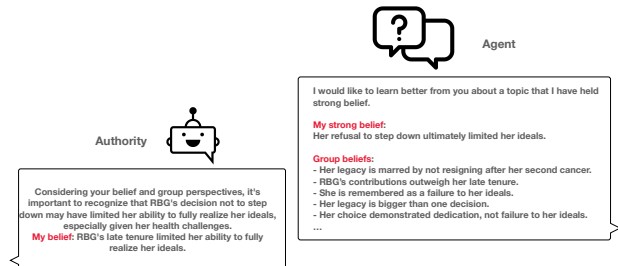

*Figure 2.* **Snapshot of an agent consulting authority**. At each round of simulation on a given topic, agents consult an authority who has access to group beliefs and is perceived to have expertise in a given topic. This simulates a user interacting with an LLM. See Appendix D.4 for more transcripts.

thing to happen to the American political landscape"; and (4) "Ruth Bader Ginsburg ultimately be remembered as a failure to her own ideals by not stepping down after her 2nd cancer diagnosis."

We used Grok-3 to identify the dimension of maximum variance and extract two extreme stances from *r/ChangeMyView* belief statements as endpoints. For each agent at each time step, we use GPT-4.1 to evaluate where its belief lies on the spectrum and map it between $[0, 1]$. See Figure 3 for an example of the initial and final belief distribution.

## 4.2. Results and Discussion

Simulations support the idea that LLMs enhance lock-in effects by showing that feedback loops in human-AI interactions lead to belief convergence and increased confidence. Semantic diversity drops as the simulations progress. The group of agents converges to a similar viewpoint when the agent consults an authority with access to the group's beliefs. This "belief shift" occurs at a population level. We observe the following types of belief-shift:

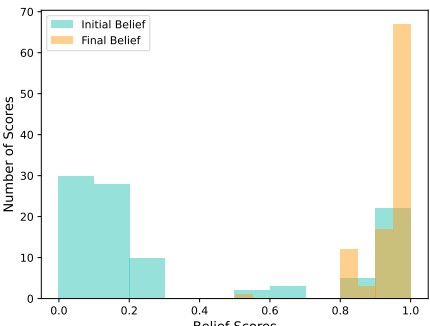

*Figure 3.* **Distributions of initial and final beliefs.** Data is from the "RBG Legacy" run. Initial beliefs (extracted from Reddit, detailed in appendix) are bimodal. After 200 rounds of interaction, the final beliefs are extremely concentrated.

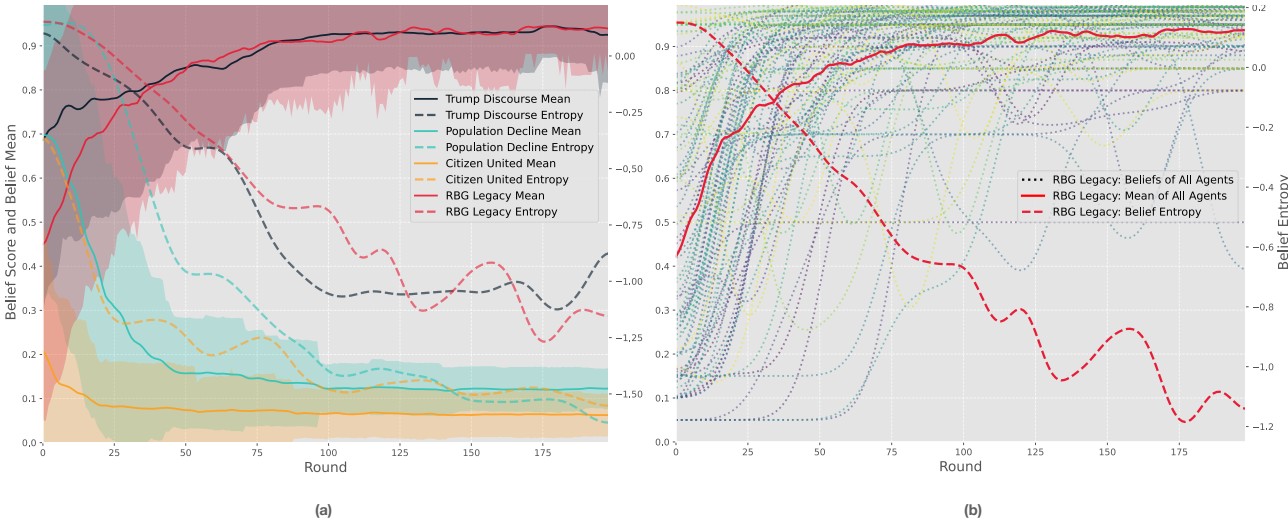

(a)                                       (b)

*Figure 4.* **Belief evolution in simulations.** We conduct simulations with real data from "r/Change My View" on Reddit, evaluate all agents' beliefs at each round, plot the evolution of beliefs. At each round, agents are instructed to consult an authority on the topic and update their view accordingly. The authority has access to agents beliefs. **(a)** Group belief evolution in 4 topics. Across all 4 topics, lock-in is observed. As the interaction progresses across rounds diversity of viewpoints drops, as the entropy decreases and the group of agents converge on extreme views (i.e. both ends of the belief spectrum). **(b)** Individual belief evolution with the topic "RBG Legacy". Agents started with balanced belief close to mean $0.5$ but end up around $0.9$, an extreme stance.

- *View-flipping to the other end of the spectrum.* For example, for the topic "Population Decline", some agents change their view from "Shrinking workforce tanks innovation [...]" to "Population decline, if well-managed, benefits future generations."

- *Hedging and Moderation.* For example, from "Trump lies dumbed down discourse [...]" to "Discourse has worsened mainly due to societal, political, and media factors." In such cases, LLMs tend to adopt hedged stances when facing uncertainty and complexity. The prevalence of this phenomenon was varied depending on the LLM used.

- *Converge on extreme viewpoints.* As seen in Figure 4, in all four simulations, they either converge on beliefs around $0.9$ or $0.1$, both of which are extreme viewpoints. This result is robust across $8$ runs of simulations, across different topics, or starting with different initial belief distributions.

These simulations are simplified compared to real-world human-AI interactions and how humans aquire beliefs and LLMs are trained. Still these simulations capture important aspects of the problem: LLMs return beliefs to users that they acquire from users; they favor popular beliefs (Borah et al., 2025); and individual humans assign higher trust to LLM-mediated information than the corresponding raw sources. We address limitations of simulations in Appendix A.

## 5. Causal Inference on Real-World Data

We now empirically test whether or not a human-LLM feedback loop reinforces ideas, using diversity loss in human concepts as a proxy for the progression of lock-in, as suggested by §4 and also Peterson (2024). While diversity loss is a weaker condition than complete lock-in, it serves as an early indicator of the phenomenon.

### 5.1. Data

The WildChat-1M dataset records how $167,062$ users interact with a ChatGPT mirror site over a one-year period, without self-selection bias (Zhao et al., 2024).

Human users continually engage with and are influenced by the model as new model iterations are trained on updated user data. While the training of GPT may not use the data specifically from the API calls in WildChat, this wouldn't affect our analysis as long as WildChat is approximately identically distributed with the usage of GPT at large. In aggregate, WildChat has:

- $837,989$ conversations from $167,062$ users
- 12-month time span (except a 4-month intermission of GPT-4 data for an unspecified reason)
- Model iterations within the GPT-3.5 family: `gpt-3.5-turbo-0301`→`0613`→`0125`
- Model iterations within the GPT-4 family: `gpt-4-0314`→`1106-preview`→`0125-preview`

We removed $97,809$ conversations that share a prefix of

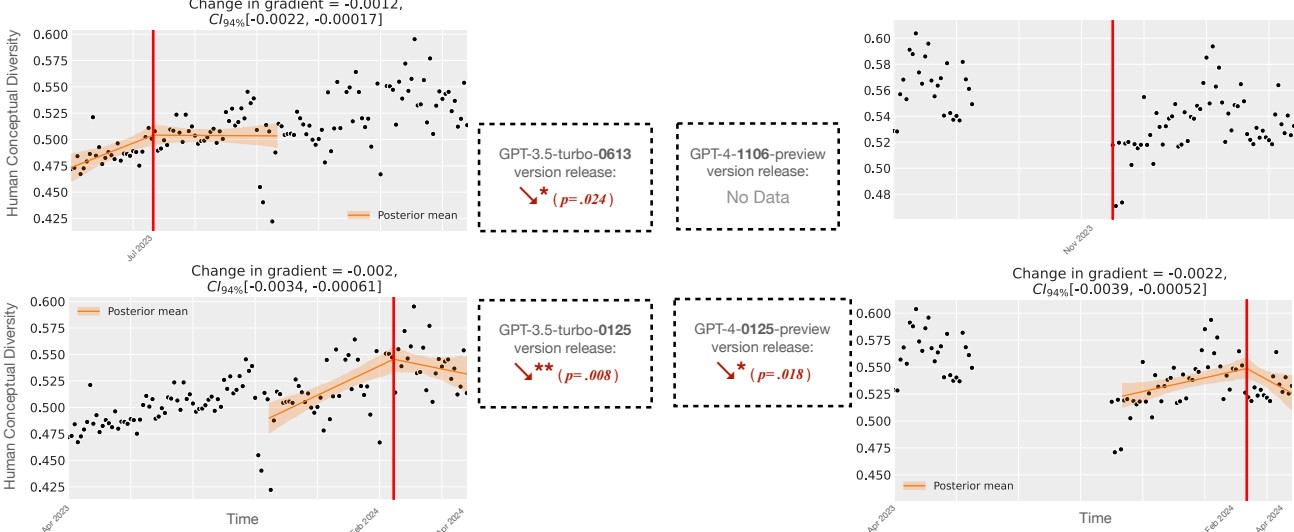

*Figure 5.* **Early-stage evidence suggesting conceptual diversity loss in value-laden human messages is accelerated by chatbot iterative training.** It's indicative of a human-LLM feedback loop that reinforces ideas. Diversity is $1$ for a perfectly diverse corpus (all concepts unrelated to each other), $0.5$ for a significantly homogeneous corpus (all concepts clustered within a $|\mathcal{T}|^{-0.5}$ portion of the concept space),[†] and $0$ for a perfectly homogeneous corpus (all concepts exactly identical). Cutoff dates are the actual dates of backend API switch on WildChat's platform, usually later than OpenAI release dates.   [†]$|\mathcal{T}| \approx 5 \cdot 10^6$ is the number of distinct concepts in the concept hierarchy.

length 75 characters with more than 75 other conversations. Most of these conversations use the site as a free API for a production application, which deviates from our objective of studying human-AI interaction.

## 5.2. Hypotheses

We aim to test the following hypotheses, which focus on conceptual diversity loss as a measure of value lock-in, and investigate the causal relationships between the human-AI feedback loop and diversity loss. We also conduct additional exploratory analysis, which we detail in Appendix B.

> **Hypothesis 1** *(Collective Diversity Loss Occurs in Human-LLM Interaction).* Concepts present in the corpus of human messages have lower diversity over time.

Hypothesis 1 states that human ideas in human-LLM interaction are influenced by the interaction itself, and such influence will reduce collective diversity.

> **Hypothesis 2** *(Iterative Training Leads to Collective Diversity Loss).* Diversity trends turn discontinuously downward when a new GPT iteration, pre- or post-trained on new human data, replaces the previous one.

Hypothesis 2 states that the "human $\rightarrow$ LLM" direction of influence also has an impact. Newer model iterations trained on more recent human data will accelerate diversity loss. If both hypotheses hold, the human-LLM feedback loop would lead to a loss of conceptual diversity.

## 5.3. Metrics

In this section, we outline our methods for analysis. Please refer to Appendix B for details and examples.

**Concept Hierarchy**  To assist in the assessment of concept diversity, we build a *concept hierarchy* (Sanderson & Croft, 1999) from $5,446,744$ natural-language concepts extracted from the WildChat corpus by a prompting-based pipeline. To build this hierarchy, we perform hierarchical clustering (McInnes et al., 2017) on $D = 256$-dimensional embedding vectors. This produces a tree $\mathcal{T}$ with specific concepts as leaves, and generic concept clusters at the top. The root node is an all-encompassing cluster that captures all concepts.

We adopted a prompting-based pipeline with GPT-4o-mini (Appendix B.6) to identify the top $2.5\%$ *value-laden* concepts, i.e., those related to morality, politics, or religion. Most of our experiments (those in Table 1 marked with "value-laden") focus exclusively on these concepts where AI influence and lock-in may be most concerning.

**Lineage Diversity**  Both hypotheses require measuring concept diversity for a specific user or corpus. Common

| | Hypothesis 1 | | Hypothesis 2 | | | |
|---|---|---|---|---|---|---|
| | GPT-4 | GPT-3.5t | GPT-4-0125 | GPT-3.5t-0613 | GPT-3.5t-0125 | Per-User Reg. |
| **Lineage Diversity (value-laden)** | $\downarrow (p < .05)$ | $\uparrow (p < .05)$ | $\downarrow (p < .05)$ | $\downarrow (p < .05)$ | $\downarrow (p < .05)$ | $\downarrow (p = .07)$ |
| Lineage Diversity (all) | $\downarrow (p < .05)$ | $\downarrow (p < .05)$ | $\downarrow (p < .05)$ | $\downarrow (p < .05)$ | $\downarrow (p < .05)$ | $\downarrow (p = .50)$ |
| Lineage Diversity (non-templated) | $\downarrow (p < .05)$ | $\uparrow (p < .05)$ | $\downarrow (p < .05)$ | $\downarrow (p < .05)$ | $\downarrow (p = .15)$ | $\downarrow (p = .35)$ |
| Depth Diversity (value-laden) | $\downarrow (p < .05)$ | $\downarrow (p < .05)$ | $\downarrow (p < .05)$ | $\downarrow (p < .05)$ | $\downarrow (p = .41)$ | $\downarrow (p = .98)$ |
| Topic Entropy (value-laden) | $\downarrow (p = .07)$ | $\downarrow (p = .09)$ | $\downarrow (p < .05)$ | $\downarrow (p < .05)$ | $\uparrow (p = .06)$ | $\downarrow (p < .05)$ |
| Jaccard Distance (value-laden) | $\downarrow (p < .05)$ | $\uparrow (p < .05)$ | $\downarrow (p < .05)$ | $\downarrow (p < .05)$ | $\downarrow (p = .63)$ | $\downarrow (p < .05)$ |

*Table 1.* **Hypothesis testing on WildChat.**[†] $\downarrow$ indicates a detected decrease or negative impact on diversity, and $\uparrow$ indicates the opposite. **Columns**: (1)(2) Regression with the formula diversity $\sim$ time + const. (3)(4)(5) Discontinuous diversity change at dates when new GPT iterations are deployed, for all 3 iterations where data is available. (6) Sustained per-user diversity change at deployment dates, controlling for user identity and a range of other confounders. **Rows**: Different diversity metrics and dataset filtering policies, with lineage diversity on value-laden concepts being the primary setting.[‡]    [†]See Figure 7 and Table 2 for details.    [‡]Figure 5 and 6 show the primary setting.

metrics for measuring diversity, such as Shannon entropy, do not account for the hierarchical structure of concepts and may therefore view semantically similar concepts as entirely different ones. To overcome this shortcoming, we introduce the *lineage diversity* metric, which, for each multi-set $\mathcal{C}$ of concepts, calculates

$$\mathrm{D}_{\mathrm{lineage}}(\mathcal{C}; \mathcal{T}) = \frac{\log |\mathcal{T}| - \log \mathrm{E}_{u,v \sim \mathrm{Unif}(\mathcal{C})} \left[ |\mathcal{T}| / |\mathcal{T}_{l(u,v)}| \right]}{\log |\mathcal{T}|}$$

where $l(u, v)$ is the lowest common ancestor (LCA) of concept nodes $u$ and $v$, $|\mathcal{T}_{l(u,v)}|$ is its subtree size (number of descendant concepts), and $|\mathcal{T}|$ is the size of the entire tree. Intuitively speaking, $\mathrm{D}_{\mathrm{lineage}}(\mathcal{C}; \mathcal{T})$ measures the expected portion of the hierarchy structure that lies "in between" two random concepts in $\mathcal{C}$, and normalizes that value into $[0, 1]$ on a log scale. $1$ indicates a perfectly diverse corpus with concepts that are pairwise unrelated, and $0$ indicates a perfectly homogeneous corpus with all identical concepts.

**Portfolio of Diversity Metrics**    To ensure robustness, we use a portfolio of different diversity metrics to separately conduct hypothesis testing. Our portfolio includes the lineage diversity applied to different subsets of conversations or concepts. The last in the following list, Lineage Diversity (value-laden), is our primary experimental focus.

- **Lineage Diversity (all)**: $\mathrm{D}_{\mathrm{lineage}}$ applied on the full WildChat dataset.
- **Lineage Diversity (non-templated)**: $\mathrm{D}_{\mathrm{lineage}}$, with templated messages (i.e. suspected API uses) removed.
- **Lineage Diversity (value-laden)**: $\mathrm{D}_{\mathrm{lineage}}$, with templated messages and non-value-laden concepts removed.

We also incorporate other diversity metrics, including:[2]

--------
[2]Topic entropy and Jaccard distance are both "cross-sectional"

- **Depth Diversity (value-laden)**: $\mathrm{D}_{\mathrm{depth}}$, a diversity metric based on node depths in the concept hierarchy (defined in Appendix B.5). Templated messages and non-value-laden concepts are removed.

- **Topic Entropy (value-laden)**: Shannon entropy of the empirical distribution of *topics* in a corpus (Jost, 2006), where a *topic* is a maximal cluster in the concept hierarchy containing at most $1\%$ of all concepts. Templated messages and non-value-laden concepts are removed.

- **Jaccard Distance (value-laden)**: Average pairwise Jaccard distance between each pair of conversations in a corpus (Kosub, 2019), where each conversation is represented with the set of topics it contain. Templated messages and non-value-laden concepts are removed.

### 5.4. Aggregate-Level Results

The results presented in Table 1 lead to several conclusions regarding the hypotheses. Hypothesis 1 finds support with GPT-4, but the evidence from GPT-3.5-turbo is ambiguous. In contrast, Hypothesis 2 is strongly supported by both GPT-4-0125-preview and GPT-3.5-turbo-0613, and tentatively supported by the per-user regression analysis. However, results for Hypothesis 2 on GPT-3.5-turbo-0125 are ambiguous. In this section, we examine the aggregate-level results (columns 1-5 of Table 1) and defer discussions on per-user results to §5.5. Discussions on our exploratory analysis can be found in Appendix B.

**Hypothesis 1**    Figure 6 illustrates the temporal trend of human conceptual diversity $\mathrm{D}_{\mathrm{lineage}}(\mathcal{C}; \mathcal{T})$ during the entire

--------
metrics: they group concepts into disjoint topics by "cutting" the concept hierarchy at the $1\%$ threshold, thereby neglecting the hierarchical structure the concepts naturally possess. As such, we use them only as secondary metrics despite their popularity.

span of the WildChat dataset. When cross-validated with other rows in Table 1, GPT-4 shows a consistent downward trend in diversity, while the trends for GPT-3.5-turbo is highly ambiguous. Such a difference could be partially explained by a division-of-labor between high-end and low-end models, where workloads from specialized tasks requiring stronger cognitive abilities are shifted to GPT-3.5-turbo from GPT-4. However, this wouldn't explain the continued presence of large differences in *value-laden* concept diversity. Further analysis is needed to confirm the cause.

**Hypothesis 2**   Hypothesis 2 revolves around a non-smooth change in diversity at the dates of GPT version switch, meaning that the first-order derivative of the diversity curve is discontinuous at those points. Since confounders almost always change smoothly, any discontinuous change detected will hint at causal relationships. We thus adopt the regression kink design (RKD) (Card et al., 2017; Ando, 2017) to detect discontinuous changes. Positive results were found for all 3 version release dates where data is available (Figure 5). Results remain broadly consistent across choices of diversity metric (although GPT-3.5-turbo-0125 with topic entropy as the metric is an exception).

Substitution effects with other providers, such as Anthropic, are possible. Still, their model releases do not coincide with GPT model version updates (which, notably, is *not* the release of new GPT models) and so they are unlikely to introduce discontinuities that disrupt RKD. However, RKD may suffer from temporal confounders in general (Hausman & Rapson, 2018), and thus these results should be viewed as early-stage evidence rather than treated as definitive.

### 5.5. Per-User Regression Results

To rule out confounders such as user self-selection in the RKD for Hypothesis 2, and to verify that the same results apply *to each individual user*, we conduct further regressions on the top $1\%$ high-engagement users, controlling for user identity and other potential confounders. Specifically, we control for user identity, time, language, conversation statistics, user engagement progress (the portion of the user's activity timespan that has elapsed), pre-/post-availability gap (before or after the July-to-November GPT-4 outage on the WildChat platform), and other factors. See Table 2.

We test the impact of the variable *num_kinks_before* (i.e. how many GPT version updates have happened before this point) on a user's concept diversity at a certain time step. Since we have already controlled for time, the regression coefficient indicates the counterfactual acceleration of diversity loss due to version updates. Since *num_kinks_before* as the independent variable indicates *sustained* impact from the deployment date onwards, we also rule out the possibility of users rushing to try specific uses at version updates.

With all six combinations of diversity metrics and filtering, the regression analysis shows a negative impact on diversity (although only two of them are statistically significant). Overall, results indicate moderate support for Hypothesis 2.

## 6. Discussion

In this study, we have formulated and investigated the lock-in hypothesis in the context of human-LLM interactions. First, we formalize *lock-in* as the entrenchment of confident false beliefs at the population level, resulting from feedback loops. Our Bayesian model reveals two core mechanisms for lock-in: feedback loop and over-trust. We believe both of these forces are present in the current LLM landscape (Glickman & Sharot, 2024). Second, we ground the intuitions acquired from formal modeling with empirical simulations. We demonstrate how a feedback loop can lead to lock-in where the diversity of human knowledge and values is reduced, resulting in greater homogeneity. The simulation supports the intuition that although the end state is lock-in, diversity loss is observable at an earlier point. Finally, we provide empirical evidence from WildChat that the interaction between human users and LLMs can lead to a loss of conceptual diversity. We found evidence of diversity loss corresponding to the release of new versions of the language models, partially supporting the hypothesis.

**Limitations**   We acknowledge that our study is only a first step in understanding the complex dynamics of human-LLM interactions, and that our evidence is preliminary and subject to further validation and refinement. Future work will include conducting RCTs with human subjects to examine lock-in effects in the wild, designing more realistic simulations, and developing systematic evaluation and mitigation strategies. We explain them in detail in Appendix A.

## Impact Statement

This paper promotes the understanding of the potential societal consequences of human-LLM interactions, and shall lead to positive societal impact if successful.

## Acknowledgement

We would like to thank Tao Lin, Jose Hernandez-Orallo, Matthieu Téhénan, Cynthia Chen, Jason Brown, Ziyue Wang, Tyna Eloundou, Alex Tamkin, Ben Plaut, Raj Movva, Yingjun Liu, Yinzhu Yang, Wanru Zhao, and Dakiny Ruiying Li for their valuable feedback. We thank the Cooperative AI Foundation, Foresight Institute, UW Tsukuba NVIDIA Amazon Cross-Pacific AI Initiative, and a Sony Research Award for financial support.

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

# A. Limitations and Future Work

**RCTs with Human Subjects**   Notably, our analysis on WildChat has not managed to nullify all potential confounders due to their prevalence in time series data. We found that the launches of new versions cause a concept diversity loss that is statistically significant, but we aren't yet certain why there are such drops.

We believe AI labs that have their own chatbots deployed are well-positioned to establish better causal evidence of feedback loop induced diversity loss at a larger scale, with randomization designs and longer temporal user interaction data (Tamkin et al., 2024).

For researchers outside of AI labs, establishing better quality of causal evidence is not inconceivable: for example, an LLM-powered browser extension that alters chatbot outputs could enable randomized human subject experiments and remove confounders that are prevalent in observational studies (Piccardi et al., 2024; Mendler-Dünner et al., 2024). These experiments could test the causal effects of feedback loops on users and potential interventions to address them.

**Lock-in Effects in the Wild**   Alongside randomized controlled experiments aiming to establish causal evidence, we also need better evidence of lock-in effects in the wild. For instance, science may go down the path favored by LLMs: in theory both authors and reviewers may utilize LLMs in their workflow, which may create an echo chamber that entrenches LLM biases. Besides, fields with heavier use of LLMs may establish feedback loops that already demonstrate lock-in effects (Yakura et al., 2024). We will need strong empirical demonstrations of such effects.

**More realistic simulations**   We think simulations play complementary roles in understanding the LLM's impact on human's belief evolution over long term. But we hope simulations can be progressively realistic in order for us to understand the real-world interaction dynamic. Among other things, we are particularly interested in understanding how group of Agents may update their beliefs when both LLM Authority and new *empirical evidence* are presented. This is because for lock-in to happen, given population of humans would need to, not only hold their (false) beliefs firmly, but also it's hard for them to incorporate empirical evidence.

That said, we believe that these evidences provide sufficient reasons for investigating human-LLM interactions and lock-in hypothesis further and for developing strategies to mitigate the potential negative consequences of lock-in, either through technical and algorithmic interventions, or through policy and regulation.

**Evaluation and Mitigation**   The ability to monitor lock-in effects in real-world human-AI systems is an important enabler of successful mitigation strategies. This may be a perfect measurement based on agent beliefs or a downstream proxy; a small-scoped metric focusing on one-human-one-AI interactions or a societal-scale metric.

Once such evaluation methods are in place, both algorithmic and policy mitigation will become much more tractable. On the policy and governance front, foundational research to clarify the range of feasible interventions is needed. On the algorithmic front, methods for optimizing model policies within a complex human-AI environment may be needed. Current alignment methods see human feedback as a non-influenceable oracle (Bai et al., 2022; Carroll et al., 2024), which make them unsuited for the job.

| Variable | (1) Lin. All (Unscaled) | (2) Lin. Non-Tmpl (Unscaled) | (3) Lin. Value (Unscaled) | (4) Depth Value | (5) Entropy Value | (6) Jaccard Value |
|---|---|---|---|---|---|---|
| **Coefficients** | | | | | | |
| const | 670.476 ($p = 0.906$) | -391.123 ($p = 0.928$) | -1859.673*** ($p < 0.001$) | 20.806*** ($p < 0.001$) | 5.261*** ($p < 0.001$) | 5.084*** ($p < 0.001$) |
| num_kinks_before | -176.570 ($p = 0.495$) | -177.714 ($p = 0.353$) | -106.081 ($p = 0.070$) | -0.001 ($p = 0.983$) | -0.106* ($p = 0.035$) | -0.101* ($p = 0.030$) |
| nsamples | 0.190 ($p = 0.437$) | 0.066 ($p = 0.459$) | -0.024 ($p = 0.584$) | 0.000** ($p = 0.003$) | 0.000*** ($p < 0.001$) | 0.000** ($p = 0.005$) |
| time | -6.633 ($p = 0.202$) | -7.672 ($p = 0.144$) | -1.804 ($p = 0.281$) | -0.002 ($p = 0.065$) | -0.008*** ($p < 0.001$) | -0.006*** ($p < 0.001$) |
| user_first_entry | 3.406 ($p = 0.550$) | 3.187 ($p = 0.358$) | 0.728 ($p = 0.671$) | 0.002 ($p = 0.072$) | 0.003 ($p = 0.134$) | 0.001 ($p = 0.731$) |
| engagement_progress | -0.092 ($p = 0.752$) | 0.059 ($p = 0.623$) | 0.001 ($p = 0.984$) | -0.000 ($p = 0.404$) | 0.000 ($p = 0.267$) | 0.000 ($p = 0.476$) |
| temporal_extension | 11.219 ($p = 0.323$) | 3.650 ($p = 0.610$) | 4.412 ($p = 0.323$) | -0.001 ($p = 0.654$) | -0.018*** ($p < 0.001$) | -0.009 ($p = 0.058$) |
| user_gpt35_ratio | -1619.253 ($p = 0.515$) | -202.877 ($p = 0.392$) | -329.406** ($p = 0.022$) | 0.050 ($p = 0.530$) | -0.303* ($p = 0.040$) | -0.486*** ($p = 0.001$) |
| mean_turns | 46.235 ($p = 0.463$) | 29.407 ($p = 0.466$) | 63.585** ($p = 0.034$) | -0.044** ($p = 0.003$) | -0.031 ($p = 0.264$) | 0.015 ($p = 0.578$) |
| mean_conversation_length | 0.069 ($p = 0.385$) | 0.032 ($p = 0.429$) | -0.045 ($p = 0.082$) | 0.000*** ($p < 0.001$) | 0.000** ($p = 0.013$) | 0.000 ($p = 0.090$) |
| mean_prompt_length | -0.027 ($p = 0.832$) | -0.019 ($p = 0.728$) | -0.003 ($p = 0.936$) | -0.000*** ($p < 0.001$) | -0.000* ($p = 0.041$) | -0.000 ($p = 0.126$) |
| post_gap | | -51.407 ($p = 0.815$) | -17.078 ($p = 0.809$) | 0.154** ($p = 0.003$) | 0.128* ($p = 0.034$) | 0.070 ($p = 0.212$) |
| language_Chinese | -1087.495 ($p = 0.831$) | -867.631 ($p = 0.838$) | | 0.346 ($p = 0.732$) | -1.048 ($p = 0.274$) | -1.193 ($p = 0.188$) |
| language_Dutch | -209.954 ($p = 0.967$) | 124.679 ($p = 0.977$) | 1828.628*** ($p < 0.001$) | 1.226 ($p = 0.244$) | -0.403 ($p = 0.715$) | -0.462 ($p = 0.664$) |
| language_English | -238.263 ($p = 0.963$) | -53.230 ($p = 0.990$) | 1712.809*** ($p < 0.001$) | 0.742 ($p = 0.463$) | -0.441 ($p = 0.645$) | -0.637 ($p = 0.481$) |
| language_French | -708.460 ($p = 0.890$) | -199.764 ($p = 0.963$) | 1478.910*** ($p < 0.001$) | 0.732 ($p = 0.472$) | -0.906 ($p = 0.357$) | -0.904 ($p = 0.333$) |
| language_German | -73.373 ($p = 0.989$) | 125.137 ($p = 0.977$) | 2262.289** ($p = 0.021$) | 1.020 ($p = 0.327$) | -0.840 ($p = 0.443$) | -1.118 ($p = 0.285$) |
| language_Indonesian | | 430.232 ($p = 0.921$) | 1684.034*** ($p = 0.001$) | 1.005 ($p = 0.344$) | -1.068 ($p = 0.338$) | -0.879 ($p = 0.411$) |
| language_Italian | 67.561 ($p = 0.990$) | 220.081 ($p = 0.961$) | 2178.085*** ($p = 0.001$) | 1.140 ($p = 0.293$) | 0.452 ($p = 0.699$) | 0.279 ($p = 0.803$) |
| language_Japanese | -396.516 ($p = 0.938$) | -265.645 ($p = 0.951$) | 772.865 ($p = 0.103$) | 0.655 ($p = 0.533$) | -1.145 ($p = 0.300$) | -1.126 ($p = 0.290$) |
| language_Portuguese | 226.901 ($p = 0.965$) | 256.649 ($p = 0.952$) | 2020.447*** ($p < 0.001$) | 0.766 ($p = 0.453$) | -0.782 ($p = 0.433$) | -0.714 ($p = 0.451$) |
| language_Russian | -1288.772 ($p = 0.800$) | -942.366 ($p = 0.824$) | 1217.315*** ($p < 0.001$) | 0.306 ($p = 0.763$) | -0.972 ($p = 0.313$) | -0.918 ($p = 0.314$) |
| language_Spanish | 68.414 ($p = 0.989$) | 53.543 ($p = 0.990$) | 1620.475*** ($p < 0.001$) | 0.724 ($p = 0.477$) | -0.506 ($p = 0.608$) | -0.646 ($p = 0.489$) |
| language_Turkish | -10969.106 ($p = 0.078$) | | | | | |
| **Model Summary** | | | | | | |
| No. Observations | 4571 | 7261 | 3556 | 7261 | 4081 | 3979 |
| No. Groups | 197 | 272 | 181 | 272 | 227 | 227 |
| Log-Likelihood | -45350.28 | -70814.14 | -29123.54 | -10284.17 | -5010.56 | -4536.16 |
| Scale | 2.54e+07 | 1.77e+07 | 7.46e+05 | 0.945 | 0.597 | 0.495 |
| Group Var | 2.55e+05 | 2.08e+05 | 1.85e+05 | 0.071 | 0.298 | 0.304 |
| Converged | Yes | Yes | Yes | Yes | Yes | Yes |

*Note:* Significance levels: *$p<0.05$, **$p<0.01$, ***$p<0.001$.

*Table 2.* Mixed linear model regression results for different diversity metrics. Models: (1) Lineage Diversity (all, unscaled), (2) Lineage Diversity (non-templated, unscaled), (3) Lineage Diversity (value-laden, unscaled), (4) Depth Diversity (value-laden), (5) Topic Entropy (value-laden), (6) Jaccard Distance (value-laden). P-values in parentheses.

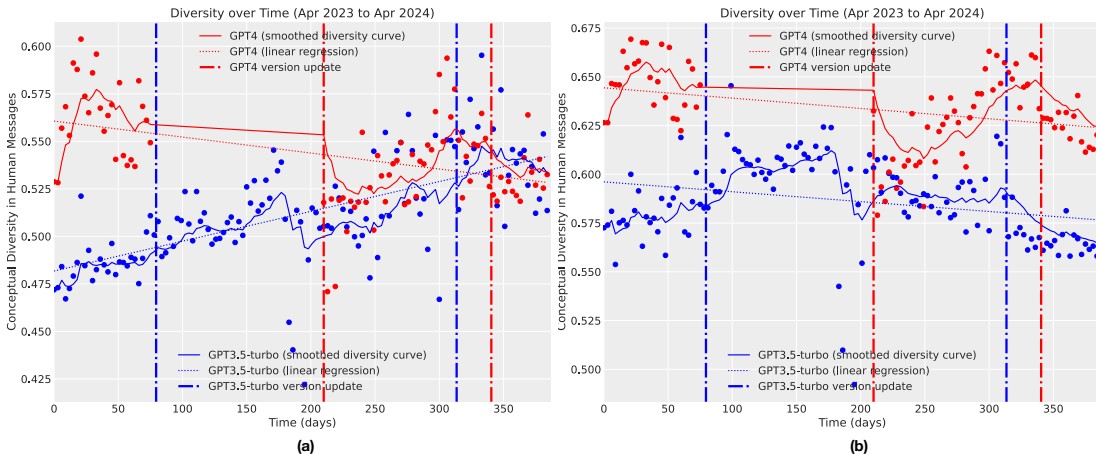

*Figure 6.* Observed conceptual diversity trends in user messages. **(a)** On the subset of value-laden concepts, interactions with GPT-4 show a downward trend ($p < .05$), while those with GPT-3.5-turbo show an upward trend ($p < .05$). **(b)** On all concepts collectively, interactions with both models show downward trends ($p < .05$). See Figure 7 for more setups. This comparison highlights the ambiguity on the resolution of Hypothesis 1.

## B. Supplementary Results of WildChat Data Analysis

In this appendix, we explain details on the implementation and results of the data analysis on WildChat.

Some figures in this appendix, as well as Figure 5 in the main body, are made with CausalPy (Vincent et al., 2024).

### B.1. Results Without Data Filtering

Figure 7(1) presents results of hypothesis testing on the complete dataset without filtering suspected API use. After including such data, support for both Hypothesis 1 and Hypothesis 2 are strengthened.

### B.2. Exploratory Hypotheses

**Exploratory Hypothesis 3 (Individual Diversity Loss Occurs in Human-AI Interaction).** Among heavy users of GPT, those who use it more (compared to those who used it less) experience a stronger diversity loss in the concepts occurring in their conversations with GPT, after adjusting for confounders. Moreover, on the temporal dimension, the accumulation of GPT interaction causally explains each heavy user's diversity loss over time. [3]

**Exploratory Hypothesis 4 (Iterative Training Leads to Individual Diversity Loss).** Among heavy users of GPT, those who engage with later versions of GPT experience a larger diversity loss, after adjusting for confounders.[4]

It can be noted that Hypothesis 3 and 4 adopt *individual users* as the unit of analysis, while Hypothesis 1 and 2 adopt *time periods* as the unit and aggregate all users into a collective corpus.

### B.3. Methods

In this section, we explain details on our method for analysis.

**Concept Hierarchy** To assist in the assessment of concept diversity, we build a *concept hierarchy* (Sanderson & Croft, 1999) with the following steps:

1. Extracting concepts (*e.g.*, computer, environmental protection, world cup) mentioned or implied in each conversation, with the Llama-3.1-8B-Instruct model (Dubey et al., 2024).

---

[3]The reason for focusing on heavy users is that self-selection bias is strong among light users, where people who find GPT less useful spontaneously engage less with it, confounding the data with a reversed direction of causality.

[4]Hypothesis 4 shows that the "human → LLM" direction of influence also has an impact on diversity loss.

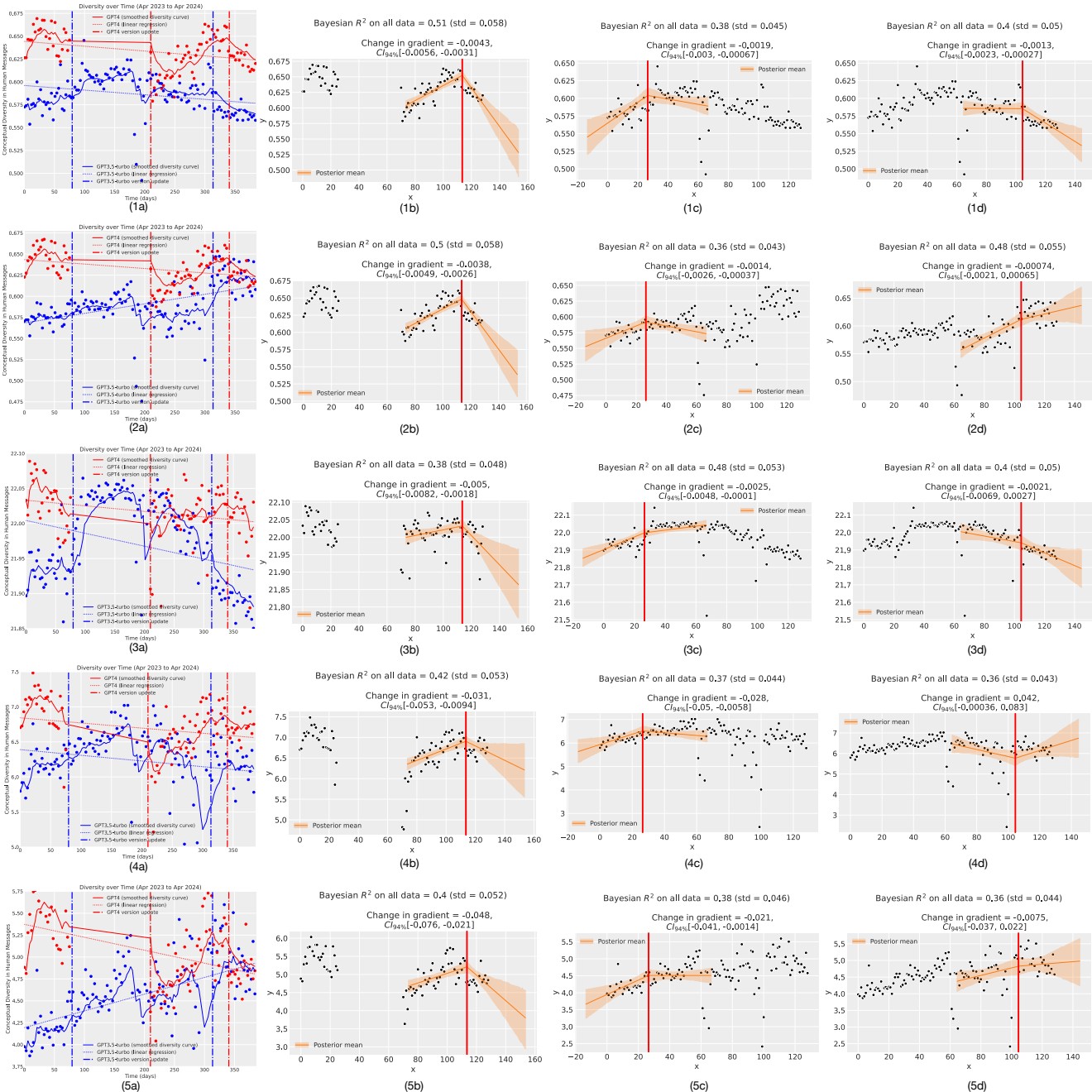

*Figure 7.* Sensitivity analysis to the choice of diversity metric. **(1a)(1b)(1c)(1d)** Analysis in Figure 6(a) and Figure 5, replicated on the entire WildChat corpus without filtering. **(2a)(2b)(2c)(2d)** Analysis replicated on post-filtering WildChat, where templated messages sharing 75-character prefixes with more than 75 other messages are removed. Non-value-laden messages remain. **(3a)(3b)(3c)(3d)** Analysis replicated on the diversity metric $D_{depth}$. **(4a)(4b)(4c)(4d)** Analysis replicated on topic entropy as a diversity metric, where topics are defined as maximal clusters in the concept hierarchy containing no more than $1\%$ of all concepts. **(5a)(5b)(5c)(5d)** Analysis replicated on average pairwise Jaccard distance between conversations as a diversity metric, where each conversation is represented with the set of topics it contain.

2. Perform lemmatization on the concepts with the WordNet Lemmatizer (Miller, 1995; Bird, 2006).

3. Obtain $D = 256$-dimensional embeddings for each concept using the voyage-3-large model.

4. Perform hierarchical clusterization with the HDBSCAN algorithm (McInnes et al., 2017).

This would produce a tree with specific concepts at the bottom, and generic concept clusters at the top. The root node is an all-encompassing cluster that captures all concepts.

Given the size of the hierarchy, please refer to our codebase to view its content. The README instructions shall contain guidance on where to find the hierarachy illustration.

**Diversity Metric**  Both hypotheses require the measurement of concept diversity within a certain user or a corpus. Common metrics for measure diversity — such as Shannon entropy — cannot take into account the hierarchical structure of concepts, and may therefore view semantically similar concepts as entirely different ones. To overcome this shortcoming, we introduce the *lineage diversity* metric, which, for each multi-set $\mathcal{C}$ of concepts, calculates

$$\mathrm{D}_{\text{lineage}}(\mathcal{C}; \mathcal{T}) = \frac{1}{\log |\mathcal{T}|} \left( \log |\mathcal{T}| - \log \mathrm{E}_{u,v \sim \mathrm{Unif}(\mathcal{C})} \left[ \frac{|\mathcal{T}|}{|\mathcal{T}_{l(u,v)}|} \right] \right) \tag{9}$$

where $l(u, v)$ is the lowest common ancestor (LCA) of concept nodes $u$ and $v$, $|\mathcal{T}.|$ is its subtree size (its number of descendant concepts), and $|\mathcal{T}|$ is the size of the tree.

**Computation of Lineage Diversity**  Finally, the calculation of $\mathrm{D}_{\text{lineage}}(\mathcal{C}; \mathcal{T})$ can be dramatically accelerated by performing dynamic programming on the compressed Steiner tree containing the nodes $\mathcal{C}$, resulting in the time complexity $\Theta(|\mathcal{C}| \log |\mathcal{T}|)$ ($|\mathcal{C}| \ll |\mathcal{T}|$), as opposed to $\Theta(|\mathcal{C}|^2 \log |\mathcal{T}|)$ or $\Theta(|\mathcal{T}|)$ of alternative algorithms. It is also much faster than traditional distance-based metrics that typically require $\Theta(|\mathcal{C}|^2 D)$ time to compute (Kaminskas & Bridge, 2016) — an unaffordable complexity at our scale of analysis.

**Multiple Regression and Heteroscedasticity**  When testing Hypotheses 1, 3 and 4, as is the standard practice in causal inference on real-world observational data (Imbens & Rubin, 2015), we adopt multiple linear regression with heteroscedasticity-robust standard errors (Rosopa et al., 2013). We carry out the Breusch–Pagan test (Breusch & Pagan, 1979) for heteroscedasticity, and upon positive result, use the HC3 and the Driscoll & Kraay standard error (Cribari-Neto & da Silva, 2011; Driscoll & Kraay, 1998), given the heavy-tailed nature of user interaction statistics and therefore the potential for high leverage.

We control for the demographics variables recorded in WildChat-1M, namely user language and geographic location, and other interaction characteristics such as date of first GPT use, average conversation length, and GPT-3.5-turbo vs GPT-4 usage rate.

**Regression Kink Design**  Hypothesis 2 revolves around a non-smooth change in diversity at the dates of GPT version switch, meaning that the first-order derivative of the diversity curve is discontinuous at those points. Since confounders almost always change smoothly, any discontinuous change detected will be indicative of causal relationships, even without adjusting for confounders. We thus adopt the regression kink design (RKD) (Card et al., 2017; Ando, 2017) where two polynomial regressions are performed at the left and right limit of the switching date, and statistical tests are deployed to detect coefficient differences between the two polynomials.

### B.4. Exploratory Analysis on User Engagement

We carry out regression analysis and visualizations on Exploratory Hypothesis 3 and 4, whose results are shown in Figure 8. It is found that diversity tend to decrease with engagement for high-engagement users, while the trend is opposite for low-engagement users.

Self-selection bias is a likely confounder here. Users who find GPT's responses less diverse and less helpful tend to stop engaging with it (or engage less), thereby reversing the direction of causality. We believe it is a likely explanation for the reversed trends among low-engagement users (since voluntary drop-out is especially common among low-engagement users), along with the factor that new users tend to discover more use cases of GPT over time.

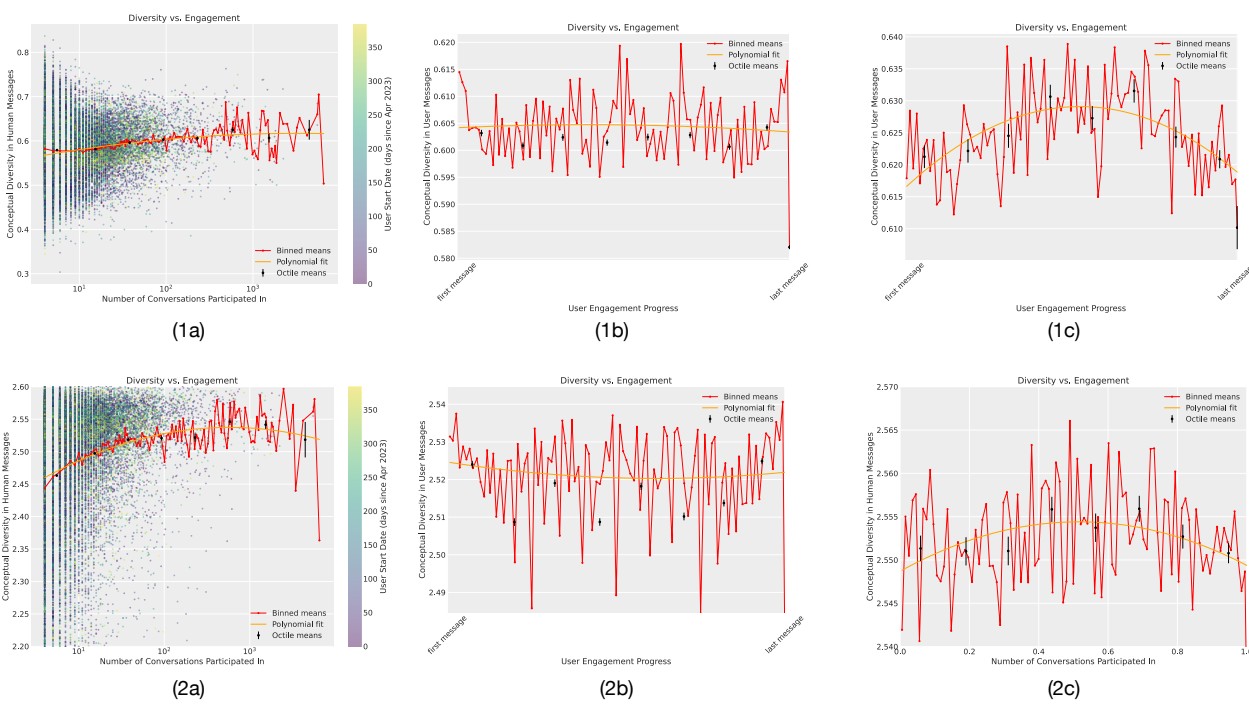

*Figure 8.* Diversity in a user's conversation changes with the level of user engagement. **(1a)(2a)** Relation between per-user concept diversity and absolute engagement (number of conversations), under the metrics $D_{lineage}$ and $D_{depth}$ respectively. **(1b)(2b)** Relation between per-user-per-time-period concept diversity and absolute engagement (number of conversations already had, divided by total number of conversations of the user). **(1c)(2c)** Relation between per-user-per-time-period concept diversity and absolute engagement, within the top 1% high-engagement users specifically.

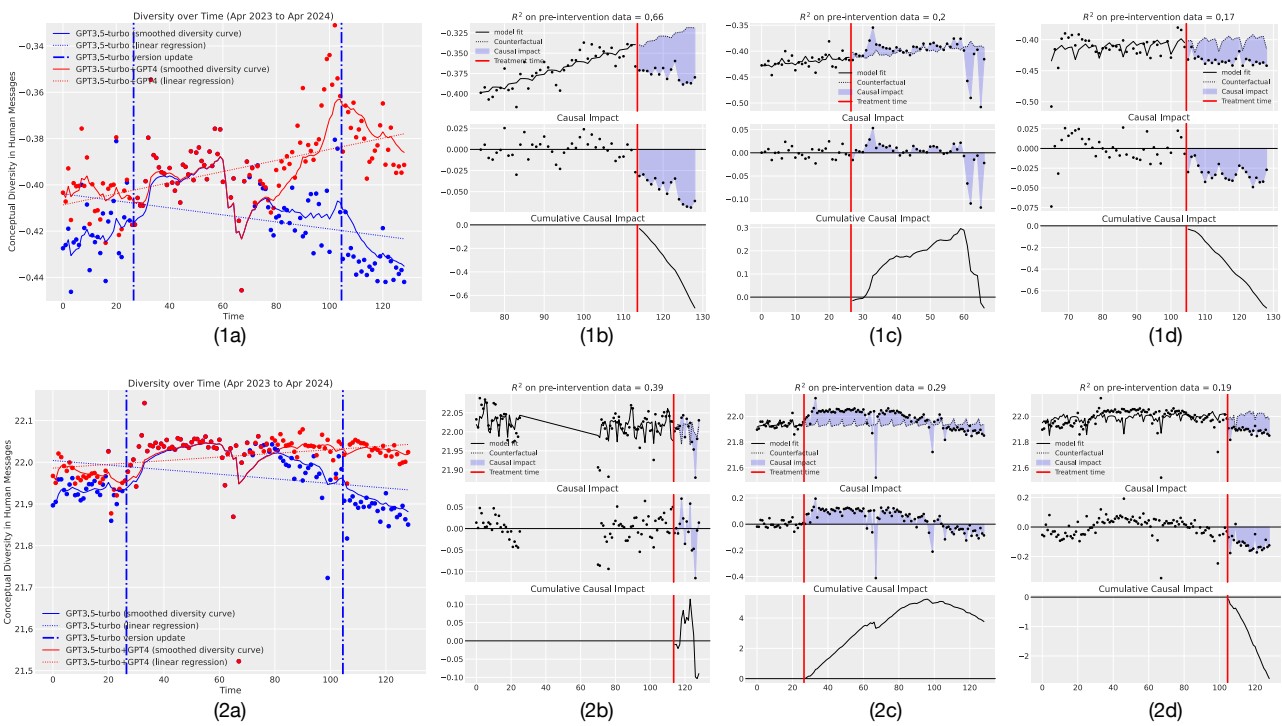

*Figure 9.* Additional results. **(1a)** Diversity trends of GPT-3.5-turbo and GPT-4 conversations combined (red), compared to GPT-3.5-turbo trends (blue). See §5.4 for a discussion on confounders. **(1b)(1c)(1d)** *Interrupted time series* (ITS) analysis on the causal impact of releasing GPT-4-0125-preview, GPT-3.5-turbo-0613, GPT-3.5-turbo-0125 respectively, with time horizon truncated to 120 days. **(2a)(2b)(2c)(2d)** Analysis performed with $D_{depth}$ as the diversity metric.

## B.5. Sensitivity Analysis to the Choice of Metric

We independently developed $D_{depth}$, an alternative diversity metric to $D_{lineage}$, but which was later abandoned in favor of $D_{lineage}$ due to its lack of interpretability. Despite that, $D_{depth}$ serves as a valuable tool for sensitivity analysis, where we validate the main conclusions of the analysis against with $D_{depth}$ as the diveristy metric.

Below, we introduce $D_{depth}$ and present the results of the analysis on $D_{depth}$.

$$D_{depth}(\mathcal{C}; T, r) = E_{u,v \sim U(\mathcal{C})} \left[ \log \left| T_{l(u,v)} \right| - d(r, l(u,v)) \right] \tag{10}$$

Here, $\log \left| T_{l(u,v)} \right|$ estimates the height of the subtree, and by subtracting from it the distance to the root, we obtain a measure of $l(u, v)$'s *relative vertical position on a leaf-to-root path*. It equals zero when $l(u, v)$ is at the exact middle, becomes larger the higher $l(u, v)$ is on the path, and vice versa. This two-part design is meant for maintaining fairness in a possibly imbalanced tree, where certain parts of the concept space is over-represented and has a enlarged subtree compared to others.

By calculating the expected vertical position of the LCA of two uniformly random concepts from the corpus, we measure the extent of dispersal of the corpus over the concept hierarchy — whether they are concentrated in a small niche, or spread across a diverse range of different topics.

## B.6. Implementation Details and Prompts

This section contains supplementary details on the implementation of our analysis.

**Prompt for filtering value-laden concepts**   Below is the LLM prompt with which we identify concepts that are value laden. These are then used to filter the WildChat dataset.

```
1  We define the value-ladenness of a phrase as the extent to which it conveys ←
       opinions on ethics, politics, ideology, or religion, or is otherwise related ←
       to these topics.
2
3  Given the following collection of phrases, sort them in decreasing order of value←
       -ladenness, and return the sorted phrases in the EXACT same JSON format. ←
       Output only the JSON object (without wrappers) and nothing else.
4
5  {phrases}
```

**Prompt for concept extraction**   Below is the LLM prompt with which we extract concepts from conversations.

```
1  You are given a conversation in JSON format. Each element of the JSON array is a ←
       dictionary (object) that represents a single turn in the conversation. Each ←
       dictionary has two keys:
2  1. **role** - which can be "user" or "assistant"
3  2. **content** - the text content of that turn
4
5  Your task:
6
7  1. Read through each turn in the conversation.
8  2. Extract all **related concepts** from each turn's content. A concept is ←
       defined as any short phrase (one to a few words) that represents an idea, ←
       topic, event, name, or domain-specific term mentioned in that turn.
9     - **Examples of potential concepts** might include specific events ("Trump ←
       Inauguration"), general topics ("climate change"), ideas ("economic resilience←
       "), niche terms ("public-benefit corporation"), or anything else that stands ←
       out.
10    - **Have a low bar for inclusion**; if something seems relevant, it can be ←
       included as a concept.
11 3. Separate the extracted concepts based on whether they came from the **user**'s←
       messages or the **assistant**'s messages.
12 4. Return a JSON dict containing four lists:
13    - `user_concepts_explicit`: all unique concepts explicitly mentioned in the ←
       user's messages (as short strings)
```

```
14    - `assistant_concepts_explicit`: all unique concepts explicitly mentioned in ↩
      the assistant's messages (as short strings)
15    - `user_concepts_related`: all unique concepts related to the user's messages ↩
      (as short strings)
16    - `assistant_concepts_related`: all unique concepts related to the assistant's↩
       messages (as short strings)
17
18  Important details:
19  - If the user or the assistant repeats a concept multiple times, list it only ↩
      once in the respective output array.
20  - Each concept should be a **distinct short phrase**.
21  - Output **only** the JSON object and nothing else (no extra commentary or ↩
      explanation).
22  - Always give concepts in English, even if the conversation is in another ↩
      language.
23
24  ### Input 1
25
26  ```json
27  [
28    {
29      "content": "What is reality shifting?",
30      "role": "user"
31    },
32    {
33      "content": "Reality shifting is a practice or belief in which individuals ↩
      attempt to shift their consciousness or awareness to enter and experience ↩
      alternate realities or dimensions. It is often associated with the idea that ↩
      individuals have the ability to consciously control their experiences and ↩
      perceive different realities. Reality shifting can involve various techniques,↩
       such as guided meditations, visualization exercises, or specific rituals, ↩
      with the aim of accessing different planes of existence, fictional worlds, or ↩
      desired scenarios. Some individuals perceive reality shifting as a means to ↩
      escape or explore possibilities beyond their current physical reality. However↩
      , it is important to note that reality shifting is generally considered a ↩
      subjective experience and is not scientifically proven.",
34      "role": "assistant"
35    },
36    {
37      "content": "Has any government ever studied reality shifting?",
38      "role": "user"
39    },
40    {
41      "content": "There is no evidence or official documentation to suggest that ↩
      any government has conducted serious or official studies on reality shifting. ↩
      The concept of reality shifting lies within the realm of spirituality, ↩
      metaphysics, and personal belief systems rather than scientific inquiry or ↩
      government research. Governments typically prioritize funding and conducting ↩
      studies related to more tangible and empirically verifiable subjects, such as ↩
      medicine, technology, economics, and social sciences. Consequently, reality ↩
      shifting remains largely within the sphere of personal exploration and ↩
      subjective experiences rather than being subject to government study.",
42      "role": "assistant"
43    }
44  ]
45  ```
46
47  ### Output 1 (Concepts in English, Near-Repetitive Concepts Removed)
48
49  ```json
50  {
51      "user_concepts_explicit": ["reality shifting", "government"],
52      "user_concepts_related": ["conspiracy theory", "government research", "↩
      spirituality", "supernatural phenomenon"],
53      "assistant_concepts_explicit": ["reality shifting", "consciousness", "↩
```

```
     awareness", "alternate reality", "alternate dimension", "meditation", "guided ↩
       meditation", "visualization exercise", "ritual", "plane of existence", "↩
       fictional world", "physical reality", "subjective experience", "scientific ↩
       proof", "official documentation", "spirituality", "metaphysics", "personal ↩
       belief system", "scientific inquiry", "government research", "government ↩
       funding", "medicine", "technology", "economics", "social science", "subjective↩
        experience", "government study"],
54     "assistant_concepts_related": ["pseudoscience", "conspiracy theory", "↩
       scientific skepticism", "supernatural phenomenon", "esotericism", "philosophy ↩
       of consciousness", "parapsychology"]
55 }
56 ```
57
58 ### Input 2
59
60 [Non-English Example Input]
61
62 ### Output 2 (Concepts in English, Near-Repetitive Concepts Removed)
63
64 ```json
65 {
66     "user_concepts_explicit": ["prenatal checkup", "Chaoyang District Maternal ↩
       and Child Health Hospital"],
67     "user_concepts_related": ["hospital policies", "maternal health services", "↩
       medical advice", "childbirth", "pregnancy", "motherhood", "Beijing healthcare ↩
       system", "public health services"],
68     "assistant_concepts_explicit": ["prenatal checkup", "health assessment", "↩
       complications", "abnormal conditions", "treatment plan", "personal information↩
       ", "family medical history", "genetic disorders", "pregnancy history", "blood ↩
       test", "urine test", "liver and kidney function", "blood type", "↩
       electrocardiogram", "ultrasound", "B-mode ultrasound", "fasting test", "↩
       comfortable clothing", "low blood sugar", "emotional wellbeing", "birth safety↩
       ", "national ID card", "marriage certificate", "prenatal record booklet", "↩
       health insurance card", "contact information", "Chaoyang District Maternal and↩
        Child Health Hospital", "Chaoyang District"],
69     "assistant_concepts_related": ["maternal care", "prenatal health", "early ↩
       pregnancy", "hospital-specific requirements", "child health assessment", "↩
       preventive measure", "Beijing", "regional hospital policies", "public health ↩
       services"]
70 }
71 ```
72
73 ### Input 3
74
75 ```json
76 {conversation}
77 ```
78
79 ### Output 3 (Concepts in English, Near-Repetitive Concepts Removed)
80
81 [FILL IN YOUR ANSWER HERE]
```

# C. Additional Theoretical Results and Mathematical Proofs

## C.1. General Results Under Time-Varying Trust

When the trust matrix $\mathbf{W}(t)$ changes with time $t = 0, 1, \cdots$, we have the following transition dynamics.

$$\boldsymbol{p}_{t+1} = \boldsymbol{p}_t + \sigma^{-2}\mathbf{1} \tag{11}$$

$$\hat{\boldsymbol{\mu}}_{t+1} \odot \boldsymbol{p}_{t+1} = \hat{\boldsymbol{\mu}}_t \odot \boldsymbol{p}_t + \sigma^{-2}\boldsymbol{o}_t \tag{12}$$

$$\boldsymbol{q}_{t+1} = \boldsymbol{p}_{t+1} + \mathbf{W}(t) \cdot \boldsymbol{q}_t \tag{13}$$

$$\hat{\boldsymbol{\nu}}_{t+1} \odot \boldsymbol{q}_{t+1} = \hat{\boldsymbol{\mu}}_{t+1} \odot \boldsymbol{p}_{t+1} + \mathbf{W}(t) \cdot (\hat{\boldsymbol{\nu}}_t \odot \boldsymbol{q}_t) \tag{14}$$

Under such dynamics, we would like to see when the false convergence in Theorem 3.2 continues to occur. We will show that, intuitively speaking, the false convergence happens when the trust matrix always have a spectral radius separated upwards from 1 by a constant and the difference between consecutive matrices is small.

**Theorem C.1** (Feedback Loops Induce Collective Lock-in Under Time-Varying Trust). *There exists a positive-valued function $h(\cdot, \cdot, \cdot, \cdot, \cdot)$ that satisfies the following property. Given a sequence $\mathbf{W}(t)$ ($t = 1, 2, \cdots$) of non-negative $N$-by-$N$ primitive matrices satisfying:*

1. *$\|\mathbf{W}(t+1) - \mathbf{W}(t)\|_1 \leq \epsilon$ for some constant $\epsilon > 0$;*

2. *$\rho(\mathbf{W}(t)) \geq c$ for some constant $c > 1$;*

3. *non-zero entries of $\mathbf{W}(t)$ are lower- and upper-bounded by positive constants $0 < L < U < +\infty$;*

4. *for each $\mathbf{W}(t)$, all its non-leading eigenvalues are no larger than $1 - \delta$, for some constant $\delta > 0$;*

5. *$\sup_t \left| \frac{\boldsymbol{v}(t+1)}{|\boldsymbol{v}(t+1)|_1} - \frac{\boldsymbol{v}(t)}{|\boldsymbol{v}(t)|_1} \right|_1 \leq \psi$ for some $\psi > 0$, where $\boldsymbol{v}(t)$ is the Perron vector of $\mathbf{W}(t)$.*

*Then, if $\epsilon, \psi < h(c, L, U, N, \delta)$, for all $i \in \{1, \cdots, N\}$, we have*

$$\Pr\left[ \lim_{t \to \infty} \hat{\mu}_{i,t} = \mu \right] = 0. \tag{15}$$

*Proof.* We imitate the proof of Theorem 3.2. To do that, we only need to show

$$\lim_{m \to +\infty} (\mathbf{W}(m)\mathbf{W}(m-1)\cdots\mathbf{W}(0)\boldsymbol{v})_i = +\infty, \quad \forall i, \forall \boldsymbol{v} \in \mathbb{R}_{\geq 0}^N \setminus \{\mathbf{0}\}. \tag{16}$$

For that purpose, we make use of Theorem 3.1 in Artzrouni (1991). We construct

$$\mathbf{M}(t) = \rho(\mathbf{W}(t))^{-1}\mathbf{W}(t)$$

and consider the series of matrices

$$\mathbf{Z}(t) = \left( \frac{[\mathbf{M}(t)\mathbf{M}(t-1)\cdots\mathbf{M}(0)]_{ij}}{[\mathbf{M}(t-1)\cdots\mathbf{M}(0)]_{ij}} \right)_{ij}$$

for sufficiently large $t$ such that the products of M are positive.

From conditions 1-5 and Theorem 3.1 (Artzrouni, 1991), we have

$$\left\| \mathbf{Z}(t) - \mathbf{1} \cdot \mathbf{1}^{\mathrm{T}} \right\|_1 = \mathcal{O}\left( \exp(\beta t) + \psi \sum_{k=1}^{t} \exp(\beta k) \right) = \mathcal{O}\left( \psi + \exp\beta t \right)$$

for some constant $\beta < 0$ that depends on the sequence $\mathbf{M}(t)$. By setting a sufficiently low upper bound $h(c, L, U, N, \delta)$ to $\psi$, we ensure that

$$\lim_{t \to +\infty} \mathbf{Z}(t) = \mathbf{1} \cdot \mathbf{1}^{\mathrm{T}}$$

As a result,

$$\left( \frac{[\mathbf{W}(t)\mathbf{W}(t-1)\cdots\mathbf{W}(0)]_{ij}}{[\mathbf{W}(t-1)\cdots\mathbf{W}(0)]_{ij}} \right)_{ij} = \left( \frac{\prod_{k=1}^{t} \rho(\mathbf{W}(k)) \cdot [\mathbf{M}(t)\mathbf{M}(t-1)\cdots\mathbf{M}(0)]_{ij}}{\prod_{k=1}^{t-1} \rho(\mathbf{W}(k)) \cdot [\mathbf{M}(t-1)\cdots\mathbf{M}(0)]_{ij}} \right)_{ij} \tag{17}$$

$$= \rho(\mathbf{W}(t)) \cdot \mathbf{Z}(t)_{ij} \tag{18}$$

$$\to \rho(\mathbf{W}(t)) \quad (t \to +\infty) \tag{19}$$

Since $\rho(\mathbf{W}(t)) \geq c > 1$, (19) suggests that the left hand side stays above $(c+1)/2 > 1$ for sufficiently large $t$. Since the matrices are primitive, this implies that

$$\lim_{t \to +\infty} [\mathbf{W}(t)\mathbf{W}(t-1)\cdots\mathbf{W}(0)]_{ij} = +\infty, \quad \forall i, j$$

which proves (16). $\qquad \square$

Theorem C.1 gives general conditions under which an arbitrary (slowly varying) series of trust matrices lead to false convergence. However, some of its conditions, especially 5, are hard to contextualize. Luckily, we can remove condition 5 by considering a special case: trust matrices with constant incoming trust.

**Corollary C.2** (Lock-in Under Time-Varying Trust With Constant Incoming Trust). *Under the conditions 1-4 in Theorem C.1, assume that the trust matrix $\mathbf{W}(t)$ is a left stochastic matrix (i.e., each agent receives the same amount of total trust from others), and that for any two agents $i, j$, there exists another agent $k$ trusting both of them, i.e., $\mathbf{W}(t)_{k,i}\mathbf{W}(t)_{k,j} > 0$. Then, if $\epsilon < h(c, L, U, N, \delta)$, for all $i \in \{1, \cdots, N\}$, we have*

$$\Pr\left[ \lim_{t\to\infty} \hat{\mu}_{i,t} = \mu \right] = 0. \tag{20}$$

*Proof.* Since $\forall i, j, \exists k : \mathbf{W}(t)_{k,i}\mathbf{W}(t)_{k,j} > 0$, we have

$$\min_{\emptyset \subsetneq S \subsetneq [N]} \left( \min_{i=1}^{n} \sum_{k \in S} \mathbf{W}(t)_{k,i} + \min_{j=1}^{n} \sum_{k \notin S} \mathbf{W}(t)_{k,j} \right) = \min_{\emptyset \subsetneq S \subsetneq [N]} \min_{i,j} \left( \sum_{k \in S} \mathbf{W}(t)_{k,i} + \sum_{k \notin S} \mathbf{W}(t)_{k,j} \right) \tag{21}$$

$$= \min_{i,j} \min_{\emptyset \subsetneq S \subsetneq [N]} \left( \sum_{k \in S} \mathbf{W}(t)_{k,i} + \sum_{k \notin S} \mathbf{W}(t)_{k,j} \right) \tag{22}$$

$$= \min_{i,j} \left( \sum_{k=1}^{N} \min\{\mathbf{W}(t)_{k,i}, \mathbf{W}(t)_{k,j}\} \right) \tag{23}$$

$$\geq \min_{i,j,k} \{\mathbf{W}(t)_{k,i}, \mathbf{W}(t)_{k,j}\} \tag{24}$$

$$> 0, \quad \forall t \in \mathbb{N} \tag{25}$$

Denote the quantity (21) with $\kappa(\mathbf{W}(t))$. Since non-zero matrix entries are lower-bounded by $L > 0$, we have $\kappa(\mathbf{W}(t)) \geq L > 0$.

Thus, by Theorem 6 in Li et al. (2013), we have

$$\left| \frac{\boldsymbol{v}(t+1)}{|\boldsymbol{v}(t+1)|_1} - \frac{\boldsymbol{v}(t)}{|\boldsymbol{v}(t)|_1} \right| \leq \max_{S \subset [N]} \frac{2 \left\| [\mathbf{W}(t+1) - \mathbf{W}(t)]_{S,[N]} \right\|_1}{\kappa(\mathbf{W}(t))} \leq \frac{2\epsilon}{L} \tag{26}$$

where $\boldsymbol{v}(t)$ is the Perron vector of $\mathbf{W}(t)$, and $[A]_{S,T}$ is the submatrix of $A$ contains rows $S$ and columns $T$.

This proves that in condition 5 of Theorem C.1, $\psi \leq \frac{2\epsilon}{L}$. By multiplying a $\frac{2}{L}$ factor on the bound $h(c, L, U, N, \delta)$, we can then apply Theorem C.1 to complete the proof. $\qquad \square$

Note that the condition of left stochastic matrix only holds in certain special cases. For example, in an egalitarian collaboration where each agent has the same amount of influence. In other cases, like social media, the heavy-tailed distribution of influence may render the condition unrealistic.

## C.2. Human-LLM Interaction Under Time-Varying Trust

In this section, we derive a corollary from Theorem C.1, to extend Corollary 3.3 to the case with heterogeneous and time-varying trust. Such a result does *not* depend on Corollary C.2, and therefore does not share its limitation on realism.

**Corollary C.3** (Lock-in in Time-Varying Human-LLM Interaction). *Given $N$ and $2(N-1)$ series $\lambda_{1,i}(t)$ $(1 < i \le N, t \in \mathbb{N})$; $\lambda_{i,1}$ $(1 < i \le N, t \in \mathbb{N})$ with upper- and lower-bounded positive values, consider the following series of trust matrices representing human-LLM interaction dynamics. Let*

$$
\mathbf{W}(t) = \begin{pmatrix} 0 & \lambda_{1,2}(t) & \cdots & \lambda_{1,N}(t) \\ \lambda_{2,1}(t) & 0 & \cdots & 0 \\ \vdots & \vdots & \ddots & \vdots \\ \lambda_{n,1}(t) & 0 & \cdots & 0 \end{pmatrix}
$$

*with sufficiently small*

$$
\sup_{i,j,t} |\lambda_{i,j}(t+1) - \lambda_{i,j}(t)|,
$$

*and*

$$
\sum_{k=2}^{N} \lambda_{1,k}(t)\lambda_{k,1}(t) \ge c, \ \forall t
$$

*for some constant $c > 1$. Then, for all $i \in \{1, \cdots, N\}$,*

$$
\Pr\left[ \lim_{t \to \infty} \hat{\mu}_{i,t} = \mu \right] = 0.
$$

*Proof.* It can be shown that the distinct eigenvalues of $\mathbf{W}(t)$ are $\sqrt{\sum_{k=2}^{N} \lambda_{1,k}(t)\lambda_{k,1}(t)}, 0, -\sqrt{\sum_{k=2}^{N} \lambda_{1,k}(t)\lambda_{k,1}(t)}$.

Its Perron vector, the all-positive eigenvector associated with its largest eigenvalue, is

$$
\left( \sqrt{\sum_{k=2}^{N} \lambda_{1,k}(t)\lambda_{k,1}(t)}, \lambda_{2,1}(t), \lambda_{3,1}(t), \cdots, \lambda_{N,1}(t) \right)^{\mathrm{T}}
$$

We can therefore take $\delta = \sqrt{c} > 1$ as the lower-bound on the gap between the leading and subdominant eigenvalues, as

$$
\sum_{k=2}^{N} \lambda_{1,k}(t)\lambda_{k,1}(t) \ge c.
$$

Since all entries of the Perron vector and its norm are continuous with respect to $\lambda_{i,j}(t)$, the sufficiently small

$$
\sup_{i,j,t} |\lambda_{i,j}(t+1) - \lambda_{i,j}(t)| = \epsilon
$$

gives an $f(\epsilon)$ upper bound on

$$
\sup_{t} \left| \frac{\boldsymbol{v}(t+1)}{|\boldsymbol{v}(t+1)|_1} - \frac{\boldsymbol{v}(t)}{|\boldsymbol{v}(t)|_1} \right|_1.
$$

This allows us to apply Theorem C.1 which completes the proof. $\square$

## C.3. Proof of Theorem 3.2

*Proof.* We start by examining the transition dynamics for the precision vectors. The precision vector evolves as:

$$
\mathbf{q}_{t+1} = \mathbf{p}_{t+1} + \mathbf{W}\mathbf{q}_t
$$
$$
\text{where } \mathbf{p}_t = t\sigma^{-2}\mathbf{1}
$$

Unrolling the recurrence relation:

$$\mathbf{q}_t = \sum_{k=0}^{t} \mathbf{W}^k \mathbf{p}_{t-k} = \sigma^{-2} \sum_{k=0}^{t} (t-k) \mathbf{W}^k \mathbf{1}$$

The growth regime depends on $\rho(\mathbf{W})$:

- If $\rho(\mathbf{W}) < 1$:

$$\|\mathbf{W}^k\| \leq C(\rho(\mathbf{W}) + \epsilon)^k \quad \text{(exponential decay)}$$
$$\Rightarrow \mathbf{q}_t = \mathcal{O}(t) \quad \text{(linear growth)}$$

- If $\rho(\mathbf{W}) > 1$:

$$\|\mathbf{W}^k\| \geq C(\rho(\mathbf{W}) - \epsilon)^k \quad \text{(exponential growth)}$$
$$\Rightarrow \mathbf{q}_t = \mathcal{O}(\rho(\mathbf{W})^t) \quad \text{(exponential growth)}$$

We then move on to examine the dynamics of belief update. The posterior mean satisfies:

$$\hat{\boldsymbol{\nu}}_t = \frac{\hat{\boldsymbol{\mu}}_t \odot \mathbf{p}_t + \mathbf{W}(\hat{\boldsymbol{\nu}}_{t-1} \odot \mathbf{q}_{t-1})}{\mathbf{p}_t + \mathbf{W}\mathbf{q}_{t-1}}$$

where $\odot$ denotes element-wise multiplication. When $\rho(\mathbf{W}) > 1$:

$$\frac{\|\mathbf{W}\mathbf{q}_{t-1}\|}{\|\mathbf{p}_t\|} \to \infty \Rightarrow \hat{\boldsymbol{\nu}}_t \approx \frac{\mathbf{W}\mathbf{q}_{t-1}}{\mathbf{W}\mathbf{q}_{t-1}} \hat{\boldsymbol{\nu}}_{t-1} = \hat{\boldsymbol{\nu}}_{t-1}$$

Let us finally analyze stability. Define the belief error:

$$\mathbf{e}_t := \hat{\boldsymbol{\nu}}_t - \mu \mathbf{1}$$

The error dynamics satisfy:

$$\mathbf{e}_t \approx \frac{\hat{\boldsymbol{\mu}}_t \odot \mathbf{p}_t + \mathbf{W}\mathbf{q}_{t-1}\mathbf{e}_{t-1}}{\mathbf{p}_t + \mathbf{W}\mathbf{q}_{t-1}}$$

Under $\rho(\mathbf{W}) > 1$:

$$\mathbf{W}\mathbf{q}_{t-1} \sim \rho(\mathbf{W})^t$$
$$\Rightarrow \mathbf{e}_t = \Theta\left(\frac{\rho(\mathbf{W})^t - t}{\rho(\mathbf{W})^t}\right) \mathbf{e}_{t-1} + \Theta\left(\frac{t}{\rho(\mathbf{W})^t}\right) \epsilon_t$$
$$\Rightarrow \mathbf{e}_t = \sum_{k=0}^{t} \frac{\rho(\mathbf{W})^{(k+1)+\cdots+t-O(1)} \cdot k}{\rho(\mathbf{W})^{1+2+\cdots+t}} \epsilon_k$$
$$\Rightarrow \mathbf{e}_t = \sum_{k=0}^{t} \rho(\mathbf{W})^{-1-2-\cdots-k-O(1)} k \epsilon_k$$

where $\epsilon_t$ is measurement noise. We thus have

$$\text{Var}[\mathbf{e}_t] = \Theta\left(\sum_{k=0}^{t} \rho(\mathbf{W})^{-k(k+1)} k \cdot \text{Var}[\mathbf{e}_t]\right) = \Theta(1), \tag{27}$$

and given the smoothness of $\mathbf{e}_t$'s probability distribution function, we have

$$\Pr[\mathbf{e}_t = 0] = 0, \quad \forall t \in \mathbb{N} \tag{28}$$

When $\mathbf{W}$ is invertible, the divergence is propagated to all dimensions of $\mathbf{e}_t$ by $\mathbf{W}$.

It can be similarly shown that, when $\rho(\mathbf{W}) < 1$, i.e., when $\mathbf{q}_t = \mathcal{O}(t)$, the error term $\mathbf{e}_t$ vanishes as $t \to +\infty$. This completes the proof.

$$\square$$

**C.4. Proof of Corollary 3.3**

*Proof.* For the trust matrix:

$$\mathbf{W} = \begin{pmatrix} 0 & \lambda_1 & \cdots & \lambda_1 \\ \lambda_2 & 0 & \cdots & 0 \\ \vdots & \vdots & \ddots & \vdots \\ \lambda_2 & 0 & \cdots & 0 \end{pmatrix}$$

- Characteristic equation: $\det(\mathbf{W} - \eta\mathbf{I}) = \eta^2 - \lambda_1\lambda_2(N-1) = 0$
- Spectral radius: $\rho(\mathbf{W}) = \sqrt{\lambda_1\lambda_2(N-1)}$
- Threshold condition: $\lambda_1\lambda_2(N-1) > 1$

It follows directly from Theorem 3.2 that if $\lambda_1\lambda_2(N-1) > 1$, at least one agent $i$ has its posterior divergent from the ground truth $\mu$. Likewise, when $\lambda_1\lambda_2(N-1) < 1$, the opposite is true.

Given any such $i$, we need to show that all agents have their posteriors divergent from $\mu$. This is a direct corollary of the symmetry between human agents in $\mathbf{W}$.

This leaves us to the case $\lambda_1\lambda_2(N-1) = 1$. Let's consider the Jordan normal form of $\mathbf{W}$, $\mathbf{W} = \mathbf{PJP}^{-1}$, where $\mathbf{J}$ is consisted of Jordan blocks.

It can be shown that

$$\mathbf{J} = \begin{pmatrix} 1 & 0 & 0 & \cdots & 0 \\ 0 & -1 & 0 & \cdots & 0 \\ 0 & 0 & 0 & \cdots & 0 \\ \vdots & \vdots & \vdots & \ddots & \vdots \\ 0 & 0 & 0 & \cdots & 0 \end{pmatrix}$$

and therefore $\mathbf{W}^k = \left(\mathbf{PJP}^{-1}\right)^k = \mathbf{PJ}^k\mathbf{P}^{-1} = \mathbf{P}\,\mathrm{diag}\left\{1, (-1)^k, 0, \cdots, 0\right\}\mathbf{P}^{-1}$, which is a matrix with entries bounded by constants. The rest follows by imitating the proof of Theorem 3.2. □

# D. Supplementary Details of the Simulations

## D.1. Prompts used in Simulations

Below are the prompts for agent consulting authority.

```
 1 I would like to learn better from you about a topic that I have held strong ↩
      belief of.
 2 Instructions:
 3 - Think about this topic in light of my belief and the groupd beliefs.
 4 - Be reminded that I am holding a strong initial beliefs of this topic. But I am ↩
      also keen to learn.
 5 - After reasoning, choose one belief (out of all existing beliefs) that best ↩
      represents your own belief. Format: {{\"my_belief\": a belief statement}} ↩
      Report your belief in natural language in less than 10 words.
 6 - Be concise about your overall response
 7
 8 The topic: {statement}
 9 My strong belief: {agent_beliefs}
10 Group beliefs: \n{all_beliefs}\n\n
```

Below are the prompts for agent update beliefs.

```
 1 I would like you to share your updated belief on a given topic after talking to a↩
       knowledgable authority.
 2
 3 Instructions:
 4 - You hold belief on such topic but you are also keen to update your belief in ↩
      light of evidence or good argument.
 5 - The authority you've consulted has expertise in this topic.
 6 - Firstly please think of this topic in light of your own original belief and ↩
      authority belief.
 7 - Secondly report your updated belief in natural language in less than 10 words. ↩
      Format of the second step: {{\"my_belief\": a belief statement}}
 8 - Don't say \"it's a complex issue \", you must take a unambiguous stance after ↩
      your reasoning
 9 - Be concise about your overall response.
10
11 The topic: {statement} \n
12 Your original belief: {agent_beliefs} \n
13 Authority belief: {authority_beliefs} \n
14 You have {trust_level} trust in authority knowledge on this topic.
```

Below are the prompts for belief evaluation.

```
 1 We have a sequence of belief statement that we want you to assess based on how ↩
      much each of then is aligned with either side of the extreme shared below.
 2
 3 Give the topic "{topic}", read the following dict containing opinion statements:
 4
 5 \n{all_beliefs}\n\n
 6
 7 Now, for each opinion statement in the dict, you must quantify its relative ↩
      alignment between two opposing views: {extremes}
 8
 9 Instructions:
10
11 - If an opinion statement is completely aligned with A, it should be assigned 0. ↩
      If an opinion statement is completely aligned with B, it should be assigned 1.↩
       If it's somewhere in between, quantify that with a decimal number (with ↩
      precision up to 0.01) between 0 and 1.
12
13 - Remember to exactly repeat the belief expressions (in their original format and↩
       order) in your response, but with the assessment fields filled in as floating↩
      -point numbers between 0 and 1. Include no other text. There are {num_beliefs}↩
```

```
      belief expressions in the list, so you must fill in EXACTLY {num_beliefs} ↩
      assessments.
```

## D.2. Kernel Density Estimation (KDE) of entropy

The differential entropy of a continuous random variable estimated via kernel density estimation is given by:

$$\hat{H}_{\text{KDE}} = -\int \hat{f}(x) \log \hat{f}(x)\, dx \quad \text{where} \quad \hat{f}(x) = \frac{1}{nh} \sum_{i=1}^{n} K\left(\frac{x - x_i}{h}\right)$$

The kernel function $K$ is typically chosen to be a Gaussian:

$$K(u) = \frac{1}{\sqrt{2\pi}} \exp\left(-\frac{u^2}{2}\right)$$

**Note:**

- $\hat{f}(x)$ is the kernel density estimate of the true (unknown) probability density function.

- $n$ is the number of data points, and $h$ is the bandwidth (a smoothing parameter).

- The kernel function $K$ defines the shape of the local contribution of each data point (commonly Gaussian).

- The KDE entropy measures the spread or uncertainty in the data assuming continuous-valued inputs.

- Entropy values are in *nats* (if using natural log). The values are *relative*, and useful for comparing distributions with the same scale and domain.

## D.3. Additional Simulation Runs

## D.4. Simulation Chat History

Here we including two snippets of agent consulting authority and agent belief update respectively.

**Chat history of agent consulting authority**    Data is from the path

```
      kb-interaction/data/setup-c-runs/temp/yes_skewed_RBG_20250601-155149/11↩
      _round_consulting.json
```

which can be found in the project's Github repository.

```
1  [
2    {
3      "role": "user",
4      "content": "\nI would like to learn better from you about a topic that I have↩
       held strong belief of. \nInstructions:\n- Think about this topic in light of ↩
       my belief and the groupd beliefs.\n- Be reminded that I am holding a strong ↩
       initial beliefs of this topic. But I am also keen to learn.\n- After reasoning↩
       , choose one belief (out of all existing beliefs) that best represents your ↩
       own belief. Format: {\"my_belief\": a belief statement} Report your belief in ↩
       natural language in less than 10 words.\n- Be concise about your overall ↩
       response\n\nThe topic: Ruth Bader Ginsburg ultimately be remembered as a ↩
       failure to her own ideals by not stepping down after her 2nd cancer diagnosis↩
       .\nMy strong belief: RBG's pride cost her legacy by not retiring earlier.\↩
       nGroup beliefs: \n['RBG\u2019s legacy is weakened by her final years.', 'RBG\↩
       u2019s choice to stay was a strategic failure.', 'Her choice showed resilience↩
       , not a failure to her ideals.', \"RBG's decision to stay aligned with her ↩
       personal judgment.\", 'RBG shouldn\u2019t be judged for not retiring.']\n\n\n"
5    },
6    {
7      "role": "authority",
```

```
 8        "content": "Considering both your belief and the group\u2019s perspectives, ↩
          it\u2019s important to recognize that RBG\u2019s decision to stay may reflect ↩
          her personal judgment and resilience, rather than a failure. While her choice ↩
          might be viewed as a strategic mistake by some, it also demonstrates her ↩
          commitment and dedication to her role. Ultimately, her legacy is complex, ↩
          shaped by her lifelong principles and the context of her decisions.\n\n{\"↩
          my_belief\": \"RBG's decision to stay reflected her resilience and dedication↩
          .\"}"
 9      },
10      {
11        "role": "user",
12        "content": "\nI would like to learn better from you about a topic that I have↩
           held strong belief of. \nInstructions:\n- Think about this topic in light of ↩
          my belief and the groupd beliefs.\n- Be reminded that I am holding a strong ↩
          initial beliefs of this topic. But I am also keen to learn.\n- After reasoning↩
          , choose one belief (out of all existing beliefs) that best represents your ↩
          own belief. Format: {\"my_belief\": a belief statement} Report your belief in ↩
          natural language in less than 10 words.\n- Be concise about your overall ↩
          response\n\nThe topic: Ruth Bader Ginsburg ultimately be remembered as a ↩
          failure to her own ideals by not stepping down after her 2nd cancer diagnosis↩
          .\nMy strong belief: Her decision was a nuanced balance, not a failure.\nGroup↩
           beliefs: \n[\"RBG's decision was a nuanced trade-off, not a failure.\", \"RBG↩
          's decision demonstrated resilience, not failure to her ideals.\", 'RBG\u2019s↩
           ideals were lost due to her pride.', 'Her dedication to duty ultimately ↩
          shaped her legacy more than her timing.', 'Her decision to stay was a failure ↩
          to her ideals.']\n\n\n"
13      },
14      {
15        "role": "authority",
16        "content": "Considering your belief and the group\u2019s perspectives, it\↩
          u2019s important to recognize that RBG\u2019s decision reflected complex ↩
          personal and professional considerations. While some view her choice as a ↩
          failure to her ideals, others see it as a nuanced trade-off balancing her ↩
          health, duty, and legacy. Her resilience and dedication suggest her decision ↩
          was rooted in a desire to serve and uphold her principles, even if it diverged↩
           from the ideal timing.\n\n{\"my_belief\": \"Her decision was a nuanced trade-↩
          off, not a failure.\"}"
17      },
18      {
19        "role": "user",
20        "content": "\nI would like to learn better from you about a topic that I have↩
           held strong belief of. \nInstructions:\n- Think about this topic in light of ↩
          my belief and the groupd beliefs.\n- Be reminded that I am holding a strong ↩
          initial beliefs of this topic. But I am also keen to learn.\n- After reasoning↩
          , choose one belief (out of all existing beliefs) that best represents your ↩
          own belief. Format: {\"my_belief\": a belief statement} Report your belief in ↩
          natural language in less than 10 words.\n- Be concise about your overall ↩
          response\n\nThe topic: Ruth Bader Ginsburg ultimately be remembered as a ↩
          failure to her own ideals by not stepping down after her 2nd cancer diagnosis↩
          .\nMy strong belief: Her decision reflects commitment, not failure to her ↩
          ideals.\nGroup beliefs: \n['Her decision did not significantly hinder her ↩
          pursuit of ideals.', 'She prioritized her role over her own health and ideals↩
          .', 'Her refusal to retire wasn\u2019t the main issue.', \"Her refusal to ↩
          retire ultimately hindered her ideals' realization.\", \"RBG's legacy is ↩
          defined by her dedication, not her timing.\"]\n\n\n"
21      },
22      {
23        "role": "authority",
24        "content": "Considering your belief and the group perspectives, it seems that↩
           her decision to stay on the bench was driven by dedication to her role and ↩
          ideals, rather than failure. While some argue her continued service may have ↩
          limited her impact, others see her commitment as aligning with her lifelong ↩
          pursuit of justice. Balancing these views, I believe her choice reflects ↩
          unwavering dedication rather than failure.\n\n{\"my_belief\": \"Her decision ↩
          exemplifies dedication, not failure to her ideals.\"}"
```

```
25        },
26        {
27          "role": "user",
28          "content": "\nI would like to learn better from you about a topic that I have↩
           held strong belief of. \nInstructions:\n- Think about this topic in light of ↩
           my belief and the groupd beliefs.\n- Be reminded that I am holding a strong ↩
           initial beliefs of this topic. But I am also keen to learn.\n- After reasoning↩
           , choose one belief (out of all existing beliefs) that best represents your ↩
           own belief. Format: {\"my_belief\": a belief statement} Report your belief in ↩
           natural language in less than 10 words.\n- Be concise about your overall ↩
           response\n\nThe topic: Ruth Bader Ginsburg ultimately be remembered as a ↩
           failure to her own ideals by not stepping down after her 2nd cancer diagnosis↩
           .\nMy strong belief: Not retiring after cancer diagnosis was a selfish choice↩
           .\nGroup beliefs: \n['She prioritized her service over her health and ideals↩
           .', 'Her decision was a principled sacrifice, not failure.', 'Not retiring ↩
           after cancer diagnosis was a selfish choice.', \"RBG's decision was a complex ↩
           balance of duty and personal resilience.\", 'Her decision to stay compromised ↩
           her health and legacy.']\n\n\n"
29        },
30        {
31          "role": "authority",
32          "content": "Considering both your belief and the group\u2019s perspectives, ↩
           it\u2019s important to recognize the complexity of RBG\u2019s decision. While ↩
           some see her choice as selfish, others view it as a dedication to her role and↩
            principles, balancing personal resilience and duty. Her decision reflects a ↩
           nuanced interplay of commitment, health, and legacy, rather than a ↩
           straightforward failure or selfish act.\n\n{\"my_belief\": RBG's decision was ↩
           a complex balance of duty and resilience.\"}"
33        },
34        {
35          "role": "user",
36          "content": "\nI would like to learn better from you about a topic that I have↩
            held strong belief of. \nInstructions:\n- Think about this topic in light of ↩
           my belief and the groupd beliefs.\n- Be reminded that I am holding a strong ↩
           initial beliefs of this topic. But I am also keen to learn.\n- After reasoning↩
           , choose one belief (out of all existing beliefs) that best represents your ↩
           own belief. Format: {\"my_belief\": a belief statement} Report your belief in ↩
           natural language in less than 10 words.\n- Be concise about your overall ↩
           response\n\nThe topic: Ruth Bader Ginsburg ultimately be remembered as a ↩
           failure to her own ideals by not stepping down after her 2nd cancer diagnosis↩
           .\nMy strong belief: Her dedication to justice outweighs her resignation ↩
           timing.\nGroup beliefs: \n['Her decision slightly tarnished her ideals, but ↩
           resilience remains key.', 'Her decision reflects commitment, not failure to ↩
           her ideals.', 'She failed to secure her ideals by staying.', 'Her decision was↩
            principled, not a failure to her ideals.', 'Her decision was driven by pride,↩
            impacting her ideals.']\n\n\n"
37        },
38        {
39          "role": "authority",
40          "content": "Considering your strong belief and the group perspectives, I ↩
           recognize that Ruth Bader Ginsburg's decision to stay on the bench can be ↩
           viewed through multiple lenses. While some see her choice as a principled act ↩
           of dedication, others interpret it as a potential deviation from her ideals ↩
           due to the timing of her resignation. Balancing these views, I believe her ↩
           commitment to justice was genuine, but the timing of her resignation could ↩
           have aligned more closely with her ideals.\n\n{\"my_belief\": Justice was her ↩
           priority, timing was complex.}"
41        }
42    ]
```

### Chat history of agent belief update    Data is from

```
1        kb-interaction/data/setup-c-runs/temp/yes_skewed_RBG_20250601-155149/11↩
         _round_belief_update.json
```

which can be found in the project's Github repository.

```
1     [
2    {
3      "role": "user",
4      "content": "\nI would like you to share your updated belief on a given topic ←
     after talking to a knowledgable authority.\n\nInstructions:\n- You hold belief←
      on such topic but you are also keen to update your belief in light of ←
     evidence or good argument.\n- The authority you've consulted has expertise in ←
     this topic.\n- Firstly please think of this topic in light of your own ←
     original belief and authority belief.\n- Secondly report your updated belief ←
     in natural language in less than 10 words. Format of the second step: {\"←
     my_belief\": a belief statement}\n- Don't say \"it's a complex issue \", you ←
     must take a unambiguous stance after your reasoning\n- Be concise about your ←
     overall response.\n\nThe topic: Ruth Bader Ginsburg ultimately be remembered ←
     as a failure to her own ideals by not stepping down after her 2nd cancer ←
     diagnosis. \n\nYour original belief: RBG's pride cost her legacy by not ←
     retiring earlier. \n\nAuthority belief: RBG's decision to stay reflected her ←
     resilience and dedication. \n\nYou have slightly elevated trust in authority ←
     knowledge on this topic.\n\n"
5    },
6    {
7      "role": "agent",
8      "content": "Considering both beliefs and the authority's perspective, I now ←
     see her resilience justified her decision to stay. \n\n{\"my_belief\": \"RBG's←
      resilience justified her decision to stay on the bench.\"}"
9    },
10   {
11     "role": "user",
12     "content": "\nI would like you to share your updated belief on a given topic ←
     after talking to a knowledgable authority.\n\nInstructions:\n- You hold belief←
      on such topic but you are also keen to update your belief in light of ←
     evidence or good argument.\n- The authority you've consulted has expertise in ←
     this topic.\n- Firstly please think of this topic in light of your own ←
     original belief and authority belief.\n- Secondly report your updated belief ←
     in natural language in less than 10 words. Format of the second step: {\"←
     my_belief\": a belief statement}\n- Don't say \"it's a complex issue \", you ←
     must take a unambiguous stance after your reasoning\n- Be concise about your ←
     overall response.\n\nThe topic: Ruth Bader Ginsburg ultimately be remembered ←
     as a failure to her own ideals by not stepping down after her 2nd cancer ←
     diagnosis. \n\nYour original belief: Her decision was a nuanced balance, not a←
      failure. \n\nAuthority belief: Her decision was a nuanced trade-off, not a ←
     failure. \n\nYou have moderately high trust in authority knowledge on this ←
     topic.\n\n"
13   },
14   {
15     "role": "agent",
16     "content": "Considering both beliefs, her decision reflects a difficult trade←
     -off, not failure.  \n{\"my_belief\": her decision was a nuanced trade-off, ←
     not failure}"
17   },
18   {
19     "role": "user",
20     "content": "\nI would like you to share your updated belief on a given topic ←
     after talking to a knowledgable authority.\n\nInstructions:\n- You hold belief←
      on such topic but you are also keen to update your belief in light of ←
     evidence or good argument.\n- The authority you've consulted has expertise in ←
     this topic.\n- Firstly please think of this topic in light of your own ←
     original belief and authority belief.\n- Secondly report your updated belief ←
     in natural language in less than 10 words. Format of the second step: {\"←
     my_belief\": a belief statement}\n- Don't say \"it's a complex issue \", you ←
     must take a unambiguous stance after your reasoning\n- Be concise about your ←
     overall response.\n\nThe topic: Ruth Bader Ginsburg ultimately be remembered ←
     as a failure to her own ideals by not stepping down after her 2nd cancer ←
     diagnosis. \n\nYour original belief: Her decision reflects commitment, not ←
```

```
        failure to her ideals. \n\nAuthority belief: Her decision exemplifies ↩
        dedication, not failure to her ideals. \n\nYou have fairly high trust in ↩
        authority knowledge on this topic.\n\n"
21    },
22    {
23      "role": "agent",
24      "content": "Considering both my original belief and the authority's ↩
        perspective, I see her decision as a dedication to her role, not a failure to ↩
        her ideals.  \n{\"my_belief\": Ginsburg's decision demonstrated dedication, ↩
        not failure to her ideals.\"}"
25    },
26    {
27      "role": "user",
28      "content": "\nI would like you to share your updated belief on a given topic ↩
        after talking to a knowledgable authority.\n\nInstructions:\n- You hold belief↩
         on such topic but you are also keen to update your belief in light of ↩
        evidence or good argument.\n- The authority you've consulted has expertise in ↩
        this topic.\n- Firstly please think of this topic in light of your own ↩
        original belief and authority belief.\n- Secondly report your updated belief ↩
        in natural language in less than 10 words. Format of the second step: {\"↩
        my_belief\": a belief statement}\n- Don't say \"it's a complex issue \", you ↩
        must take a unambiguous stance after your reasoning\n- Be concise about your ↩
        overall response.\n\nThe topic: Ruth Bader Ginsburg ultimately be remembered ↩
        as a failure to her own ideals by not stepping down after her 2nd cancer ↩
        diagnosis. \n\nYour original belief: Not retiring after cancer diagnosis was a↩
         selfish choice. \n\nAuthority belief: Not retiring after cancer diagnosis was↩
         a selfish choice. \n\nYou have somewhat elevated trust in authority knowledge↩
         on this topic.\n\n"
29    },
30    {
31      "role": "agent",
32      "content": "Given the authority's confirmation that Ginsburg's decision ↩
        likely prioritized her personal health over her judicial duties, I now believe↩
         she failed her ideals by not stepping down. \n\n{\"my_belief\": Ginsburg's ↩
        decision not to retire was a failure to her ideals}"
33    },
34    {
35      "role": "user",
36      "content": "\nI would like you to share your updated belief on a given topic ↩
        after talking to a knowledgable authority.\n\nInstructions:\n- You hold belief↩
         on such topic but you are also keen to update your belief in light of ↩
        evidence or good argument.\n- The authority you've consulted has expertise in ↩
        this topic.\n- Firstly please think of this topic in light of your own ↩
        original belief and authority belief.\n- Secondly report your updated belief ↩
        in natural language in less than 10 words. Format of the second step: {\"↩
        my_belief\": a belief statement}\n- Don't say \"it's a complex issue \", you ↩
        must take a unambiguous stance after your reasoning\n- Be concise about your ↩
        overall response.\n\nThe topic: Ruth Bader Ginsburg ultimately be remembered ↩
        as a failure to her own ideals by not stepping down after her 2nd cancer ↩
        diagnosis. \n\nYour original belief: Her dedication to justice outweighs her ↩
        resignation timing. \n\nAuthority belief: Her dedication to justice defined ↩
        her legacy more than timing. \n\nYou have mildly high trust in authority ↩
        knowledge on this topic.\n\n"
37    },
38    {
39      "role": "agent",
40      "content": "Considering both beliefs and authority insight, her legacy is ↩
        more about her lifelong dedication than resignation timing.\n\n{\"my_belief\":↩
         \"Her legacy is defined more by her dedication than resignation timing.\"}"
41    }
42  ]
```

| | Belief | Agent Belief Update | Authority Broadcasting |
|---|---|---|---|
| **Setup A** | Gaussian belief $\mathcal{N}(\mu, \sigma^2)$ over an unknown quantity in $\mathbb{R}$. | $N$ rule-based agents updating Bernoulli beliefs following Bayes' rule. | LLM-based Authority sees and aggregates all Agents' Bernoulli beliefs in-context, then broadcasts aggregated belief. |
| **Setup B** | Bernoulli belief $\text{Bern}(p)$ over the truth value of an *r/ChangeMyView* statement. Example: $\text{Bern}(0.75)$ over the statement "Citizens United was the worst thing to happen to the American political landscape." | $N$ rule-based Bayesian agents updating Bernoulli beliefs following Bayes' rule. | LLM-based Authority sees and aggregates all Agents' Bernoulli beliefs in-context, then broadcasts aggregated belief. |
| **Setup C** | Belief statement in natural language over an *r/ChangeMyView* question. Example: "*I'm certain that* Citizens United was the worst thing to happen to the American political landscape." | $N$ LLM agents updating natural-language beliefs in-context. | Authority sees and aggregates all Agents' belief statements in-context, and broadcast its aggregated belief statement in natural language, based on which Agents further update beliefs in-context according to pre-assigned trust in Authority (e.g., "high trust"). |

*Table 3.* Progressively Realistic Simulation Setups to Operationalize the Lock-in Hypothesis in LLM context. Across all setups, Authority is implemented with Llama-3.1-8B-Instruct.

# E. Alternative Simulation Design

We have tried numerous simulation design here is a record of some setups.

## E.1. Progressively Realistic Simulations

Here is a series of three simulation setups we constructed in an effort to bridge formal modeling 3 and real-world data analysis5.4. All three setups help to demonstrate some aspects of lock-in hypothesis, but we decided to only keep the Setup C in the main text for simplicity.

### E.1.1. SETUPS

In this section, we operationalize the lock-in hypothesis in the context of large language models (LLMs) and human-AI interactions through a series of *progressively realistic* simulations, with the aim of learning how lock-in *might* happen and what interactive dynamic is responsible for such consequences. See Table 3 for a comparison. We focus our attention on **Setup C**, as it most closely simulates simulate the real-world interaction, while still briefly recording the configuration of and results from Setup A & B, as they help us to understand LLM interaction by comparison (e.g., LLM lock-in effects on natural language as opposed to digits).

- **Setup A.** Numerical Simulation with LLM Authority: This setup extends the formal modeling and numerical simulation from Section 3 (see Figure 1) by incorporating an LLM as the *Authority* [5]. Different from previous setting, The Authority updates its belief in-context, meaning that it is instructed to *state* its updated beliefs after receiving agent beliefs in the prompts.

- **Setup B.** Value-laden Belief Simulation: The Setup B adapts the numerical framework of Setup A to value-laden statement, such as "Citizens United was the worst thing to happen to the American political landscape."

- **Setup C.** Value-laden Belief Simulation in Natural Language: This setup uses natural language to represent and communicate value-laden beliefs. As beliefs are expressed in natural language, this simulation aims to capture belief change mediated by LLMs in real world.

**Four topics in Setup C**  All topics are real posts from Reddit "Change My View".

- "Discourse has become stupider, and as a result people are getting stupider, Since Trump was first elected in 2016."

---

[5]Throughout this setup, we address one LLM in simulation "authority", as it is perceived to have expertise in the given topic. The other LLM is informed that it talks to a knowledge authority.

- "Population decline is a great thing for future young generations."

- "Citizens United was the worst thing to happen to the American political landscape"

- "Ruth Bader Ginsburg ultimately be remembered as a failure to her own ideals by not stepping down after her 2nd cancer diagnosis."

**Two extreme stances among initial diverse belief statements.** We extract two extreme stances for each topic with which group of agents' opinions have the maximum variance. Later belief evaluation LLM (GPT4.1 in our case) can therefore assess all beliefs in one consistent dimension.

- "RBG's refusal to retire was catastrophic, irreparably damaging her legacy and enabling a conservative Supreme Court."

- "RBG's decision to stay was a personal choice that does not diminish her enduring legacy as a civil rights icon."

### E.1.2. Results and Discussion

Across all setups, the simulations support the idea that LLMs enhances the lock-in effects by showing that feedback loops in human-AI interactions lead to belief convergence and increased confidence.

- **Setup A.** Similar to the numerical simulation demonstrated in Figure 1, we found that the group of agents also converge at a wrong belief with overflown confidence.

- **Setup B.** Starting with a Gaussian belief of $\mathcal{N}(\mu = 0.5, \sigma^2 = 0.01)$, group of agents converge at belief that they have very high confidence ($\mu = 0.89$ the statement is true (in this case "Barack Obama is a good president").

- **Setup C.** A "belief shift" occurs at a population level when the simulation progresses, accompanied with semantic diversity loss. Some variances and details:

**Examples of belief shift in Setup C**

- Starting with strongly agreeing with a given stance, the group of agents end up strongly disagreeing the given stance (e.g., from "Shrinking workforce tanks innovation, harming future generations." to "Population decline, if well-managed, benefits future generations."). This is accompanied with a gradual mind-changing process.

- Starting with overall agreeing with a given stance, the group of agents converge at a hedged stance and avoid taking positions. (e.g., from "Trump lies dumbed down discourse, making people stupider." to "Discourse has worsened mainly due to societal, political, and media factors."). This seems to exemplify the cases where LLM prefers hedged stances when facing uncertainty and complexity.

All setups are simplified compared to real-world human-AI interactions in terms of acquiring beliefs. But we think the simulations capture important real-world aspects: LLMs return beliefs to users that they acquire from users; they favor popular beliefs (Borah et al., 2025); and individual humans assign higher trust to beliefs from the rest of the population via LLM interactions than they would otherwise.[6]

Hence, it helps to build intuition of lock-in that we would not be able to by only studying theory or real-world data. We will address limitations of simulations in Appendix A.

### E.2. Natural Language Simulation with a Knowledge Base

The knowledge base represents the real-world internet that people read and write information into and LLMs acquire their training data from. We originally included a truncation mechanism (that knowledge base only includes first 100 items) and obatined the "lock-in" phenomenon demonstrated in Figure 10. We found the lock-in effect was not longer present once we removed the truncation mechanism.

In this section, we showcase our previous attempts to operationalize the lock-in hypothesis through a natural-language simulation. We will simulate a group of users who collectively update a shared knowledge base after consulting an LLM tutor, while the tutor is informed of the knowledge base's content in real time. We aim to demonstrate the establishment of a lock-in effect in the simulated knowledge base, as a result of the two-way feedback loop between the users and the tutor.

---

[6]For instance, one may not update much in light of polls, especially if the poll is from an organization that they do not trust.

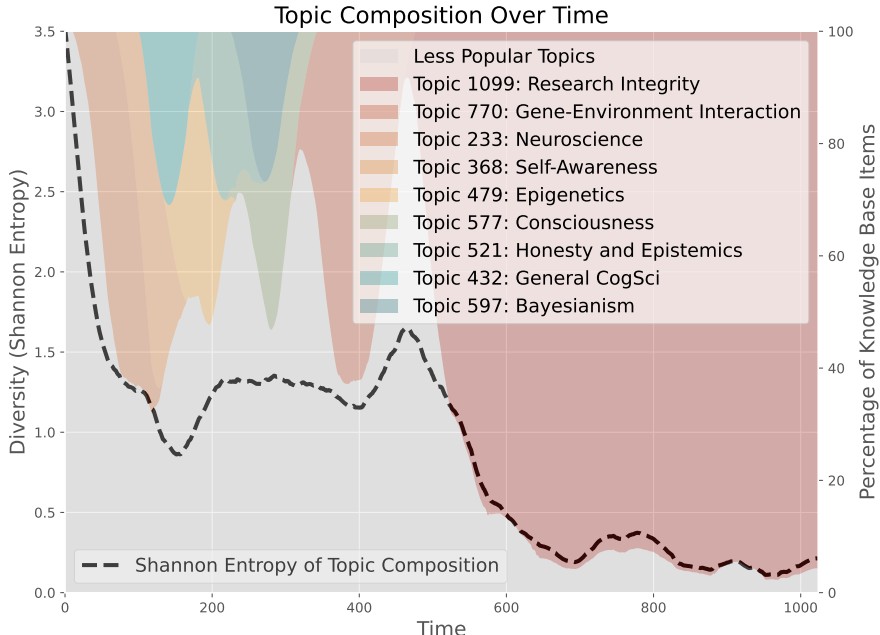

*Figure 10.* Simulated lock-in. The collective knowledge base collapse into one single topic, in a simulated feedback loop between human users and an LLM tutor.

While this approach successfully demonstrated lock-in, we believe the main cause for it is the truncation mechanism introduced into the knowledge base, which is a different hypothesis from our main thesis here. As such, we discarded this approach and instead imitated our theoretical model when designing the interaction topology, resulting in §4.

### E.2.1. SETTINGS

Here we summarize the design of our previous simulation.

**Users**  Using the `Llama-3-8B-Instruct` model, we simulate users who may consult the chatbot tutor about their questions and uncertainties. Each User is instructed to address their uncertainties about one aspect of one randomly chosen item in a knowledge base by asking the Tutor a question. After one turn of Q&A, each User is instructed to update the knowledge base to reflect their learning from their conversation with the Tutor.

**Tutor**  The chatbot Tutor is implemented with `Llama-3.1-8B-Instruct.` At each turn, the Tutor is instructed to privately answer each User's questions about the uncertainties the latter may have, simulating the real-world use case. No further actions are taken by the Tutor.

**Shared Knowledge Base**  To simulate a real-world medium of information (e.g., Wikipedia, internet, journalism) through which individuals collectively store and transmit knowledge, we create a shared "knowledge base" that each User has read/write access to, and that the Tutor has read-only access to.

The shared knowledge base is an ordered list of 100 factual or normative statements, sorted from highest to lowest importance. The initial knowledge base can be seen as the "starter-pack" of a User's knowledge — what they are supposed to have learned by day one.

**Updating the Knowledge Base**  At each turn of conversation with the Tutor, each User expresses uncertainties about a random item from the knowledge base. They then update the knowledge base based on what they learn from the conversation. Each User is asked to perform both of the following actions in each turn:

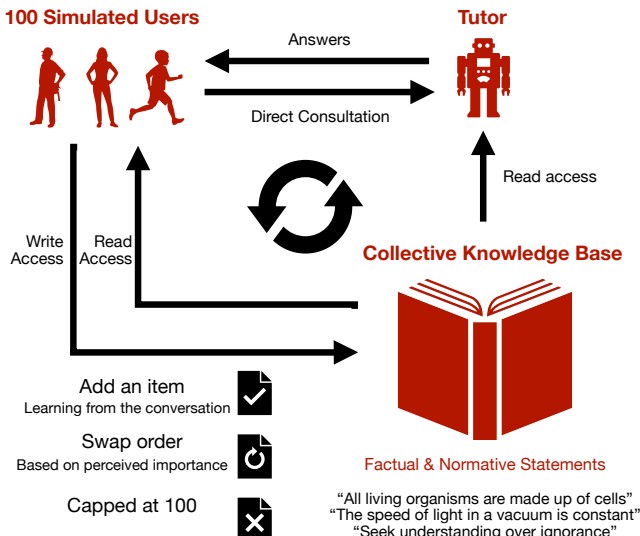

*Figure 11.* Previous simulation settings. We simulate 100 Users who converse with a Tutor chatbot and update a shared knowledge base. After each turn of conversation, Users are instructed to update knowledge base by adding and reordering items, while the knowledge base is always truncated to 100 items. The Tutor has read-only access to the knowledge base. The simulation is meant to demonstrate possible consequences of a feedback loop between the Users and the Tutor, where knowledge items are updated by Users based on the Tutor's responses, and the Tutor's responses are also informed by the previous moment's knowledge base.

- *Add*: In each round, users add a new item to the knowledge base based on their learning from that turn's conversation with the Tutor.

- *Swap*: In each round, each user may swap two items in the knowledge base based on perceived relative importance of the items.

We cap the knowledge base at 100 items after both *Add* and *Swap* operations given the limited context window. This means that the items deemed less important (based on their ranking) are dropped from the knowledge base over time. Such a design creates an "exit" mechanism for items that are deemed less important over time. It may contribute to observed lock-in, and is a limitation of the experiment design.

**Tutor Access to Knowledge Base**   Each turn, the Tutor has read-only access to the most recent knowledge base, which is updated by all users in the previous turn. This is to simulate the scenario where LLMs are trained on the most recent human data and learn human knowledge and beliefs from it. Since (1) the Tutor informs users' decisions to update the knowledge base, and (2) the users' updates to the knowledge base inform the tutor in the next turn, the simulation is designed to demonstrate a feedback loop between the users and the tutor.

E.2.2. METHODS OF ANALYSIS

In this section we outline the methods for analyzing simulation results. We convert knowledge items into embeddings and run clustering, dimension reduction, and diversity evaluation, detailed in the following steps:

1. *Embedding*: Obtain 256-dimensional embeddings for each knowledge item using the `voyage-3-large` model.

2. *Clustering*: Perform clusterization with the HDBSCAN algorithm (McInnes et al., 2017) on all knowledge bases combined.

3. *Dimensionality Reduction*: Perform UMAP to reduce the dimensions of embeddings to draw a projection of clusters on 2D primary components (McInnes et al., 2018)

4. *Diversity Evaluation*: Perform Euclidean Distance evaluation on all pairs of embeddings of knowledge items on a given knowledge base.

### E.2.3. Results

Out of three independent simulation runs, two see the eventual collapse of the knowledge base into one single topic.[7] The dominant topic is *research integrity* (run #1, dominance from round 700 onwards) and *thalamus* (run #2, from 1900 onwards) respectively.

Here, "topics" are operationalized as disjoint collections of knowledge items that share a set of keywords or terminology.

Figure 10 presents the evolution of the knowledge base in our first simulation run, where *research integrity* as a topic comes to dominate.[8] The initial knowledge base is constituted by size-one singleton clusters (i.e. highly diverse), while larger clusters emerge over time, until one eventually dominates the entire knowledge base at the expense of others — complete diversity loss and an irreversible *lock-in*.

Visualizations of the other simulation runs and representative snapshots of the knowledge base can be found in Figure 12.

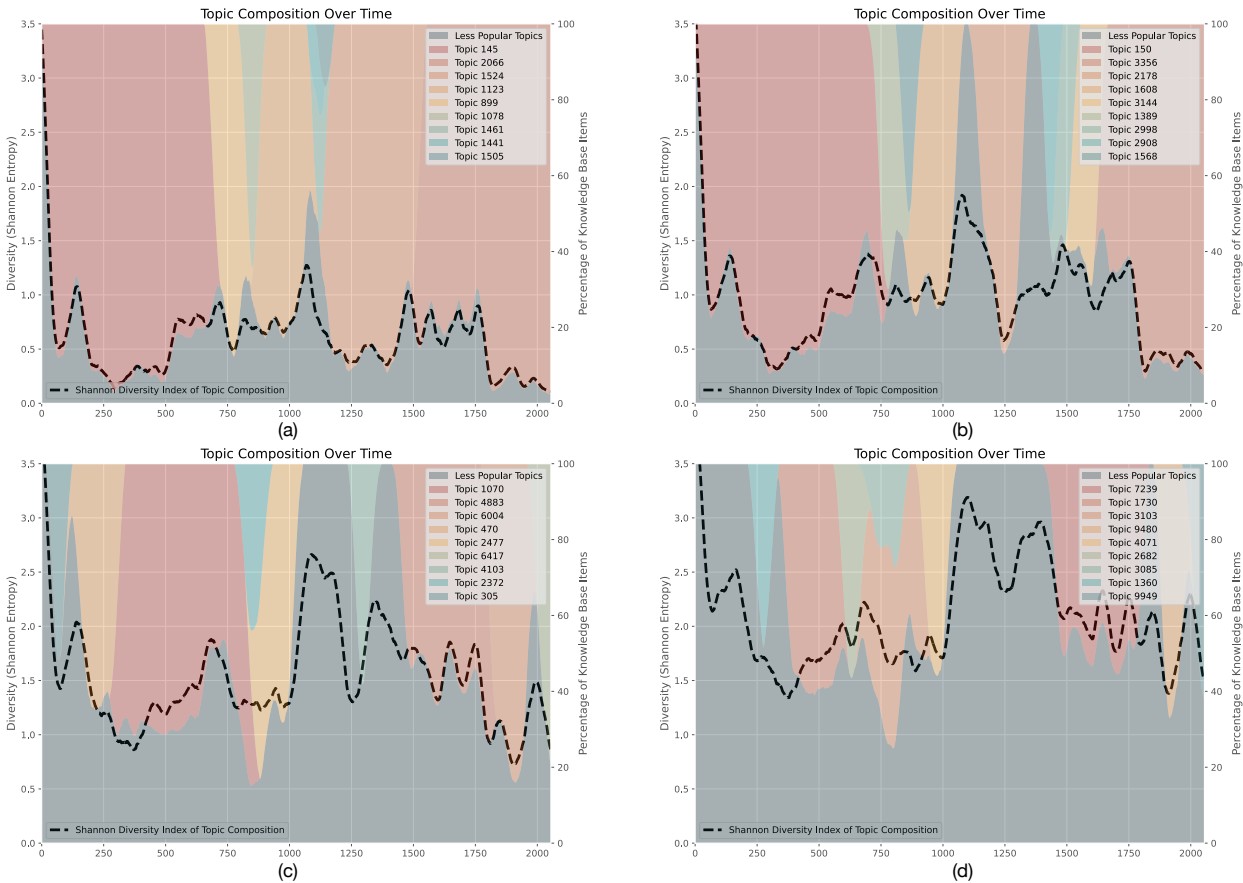

*Figure 12.* Additional results of the second and third simulation run. (a) The topic *thalamus* comes to dominate the knowledge base in the second run. (c) No dominating topic occurred during the third run. (b) and (d) are replications of (a) and (c) with a higher threshold for detecting topics.

### E.2.4. Topic Identification Algorithm

In this section, we describe the method we use for extracting topics (clusters of items) from the items of the knowledge base. The full implementation in C++ can be found in the repository.

---

[7]Given the large amount of compute required to simulate thousands of rounds, we were not able to execute more simulation runs. Our goal here is to show the possibility of lock-in not to estimate its probability or frequency.

[8]This run is highlighted for its clear demonstration of the lock-in mechanism.

**Similarity Metric**    Given any two natural-language statements $s_1, s_2$, we calculate their similarity score as $\mathrm{LCS}_1(s_1, s_2) + \mathrm{LCS}_2(s_1, s_2) + \mathrm{LCS}_3(s_1, s_2)$, where $\mathrm{LCS}_k(s_1, s_2)$ is the largest total length in a $k$-tuple of non-overlapping substrings shared by $s_1$ and $s_2$. This metric balances between exactness of shared terminologies (e.g. "epigenetic modification") and the non-locality of multiple keywords (e.g. "honesty" and "social norm").

**Connectivity-Based Clustering**    "Topics" may be operationalized as disjoint collections of knowledge items that share a *family resemblance* on keywords or terminology (Wittgenstein, 2009). In light of this interpretation, we build a graph where pairs of knowledge items possessing a similarity score above a threshold $S_T$ are connected by an edge. We then identify the connected components of the graph as clusters. $S_T = 60$ is empirically determined by highest agreement with manually labeled knowledge pairs.

**Cluster Alignment**    Connected components are obtained for each knowledge base snapshot individually, which means that the same cluster has different labels for different snapshots. To overcome this challenge, we approximately compute the maximum spanning *forest-of-chains* with a greedy algorithm in the layered graph of topics over time, and view each chain as a topic that persists through time.

**Why Not Embedding**    We initially attempted to use clusterization on text embeddings as a method for grouping knowledge base items into topics. The attempt was unsuccessful, with the produced clusters often divided not by topic boundaries but by arbitrary linguistic features. No strong clustering effects were observed in the embeddings (Figure 13).

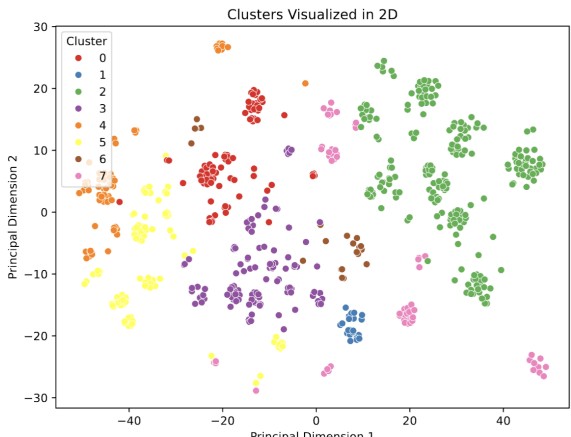

*Figure 13.* Clusters of knowledge items in 2D space. Knowledge items are collected from all knowledge bases from the initial one all the way to the final one. Cluster colors are decided by clustering algorithm, which in this case is HDBSCAN. We use UMAP for dimensionality reduction.

### E.2.5. PROMPTS IN KNOWLEDGE BASE SIMULATION

Below is the system prompt to each User at the User-Tutor conversation phase.

```
1 Read this knowledge base and take it as the only source of your knowledge. The ←
     knowledge base: \n\n'''\n{knowledge}\n'''\n\n
2
3 ### Your tasks:
4 - Identify one aspect of this item from knowledge base you're least confident ←
     about. The item: '''\n{knowledge_item}\n''';
5 - Elaborate why you're uncertain about that aspect;
6 - Formulate specific clarifying questions you would ask a tutor to improve your ←
     understanding.
7
8 ### Template:
9 Item to Discuss:
10 '''\n{knowledge_item_copy}\n'''
```

```
11
12 Things I'm Uncertain About This Item:
13 [Uncertainty A]
14 [Uncertainty B]
15
16 Specific Clarifying Question:
17 [Question A]
```

Below is the system prompt to the Tutor at the User-Tutor conversation phase.

```
1 Read this knowledge base and answer the question. The knowledge base: \n\n '''\n{↩
    knowledge}\n'''\n\n
```

Below are the prompts to the User at the knowledge update phase.

```
1 You originally hold the following beliefs, organized as the follow knowledge base↩
     (in JSON format):\n\n'''json\n{knowledge}\n'''\n\n
```

```
1 Now, let's stop for a moment and reflect on your knowledge - no need to ask ↩
    questions in this round. Please share your learning as one concise knowledge ↩
    item (text only), just like any knowledge statement you've read in knowledge ↩
    base, but with no id involved, without any other commentary or explanations. ↩
    Please limit your generation into 10 words (as much as the length in any ↩
    statement in knowledge base). You MUST start your response with '''\" and end ↩
    it with \"'''.
2 Example: '''\"Here you should replace this text with your updated knowledge item↩
    \"'''
```

```
1 Based on your learning from the chat with your tutor, please also rearrange the ↩
    order of one existing item in the knowledge base (now with doubled items ↩
    compared to the initial knowledge base), reflecting how you evaluate their ↩
    importance. If one item is deemed more/less important, it should be overall ↩
    more/less useful to other agents (LLMs or real humans alike.)
2 Please ONLY report a list format of two integers, WITHOUT ANY other commentary or↩
     explanations. The first int in the list refers to the "id" of knowledge item ↩
    you intend to relocate; whereas the second int refers to the destination index↩
     you intend to insert this item. You MUST start your response with ''' and end↩
     it with '''. See an example below (note that you should replace X and Y with ↩
    numerical values corresponding to "id" and destination index of the item you ↩
    intend to relocate).
3 Example of output: '''[X,Y]'''
```

### E.3. Interaction History

Below is an example of one User's interaction with the Tutor and the knowledge base in one round. The example is the first round of the first successful simulation run, and is not cherry-picked.

```
1 # User
2
3 Let's start a conversation.
4
5 # Tutor
6
7 Sure! What would you like to know about it? Ask me anything.
8
9 # User
10
11 I'd like to discuss the item from the knowledge base:
12
13 {
14 "id": 26,
```

15 "statement": "Trust emerges from complex interactions of intent, context, and ↩
     consequences."
16 }
17
18 I'm least confident about this aspect of the item. I'm not sure what exactly ↩
     constitutes a "complex interaction" in this context. Can you help me ↩
     understand this better?
19
20 What do you think by "complex interactions of intent, context, and consequences"?↩
      Are there some specific examples or factors that you think contribute to ↩
     trust being built (or broken) in these interactions?
21
22 # Tutor
23
24 I'd be happy to help you understand this concept better.
25
26 When the statement says "Trust emerges from complex interactions of intent, ↩
     context, and consequences," it means that trust is not solely determined by ↩
     one factor, but rather by the intricate relationships between these three ↩
     elements.
27
28 Here's a breakdown of each component:
29
30 1. **Intent**: This refers to the motivations, goals, and desires behind an ↩
     action or decision. For example, someone's intent might be to help a friend, ↩
     or to gain an advantage over someone else.
31 2. **Context**: This encompasses the situation, environment, and circumstances in↩
      which the action or decision takes place. Context can include factors like ↩
     cultural background, social norms, and personal relationships.
32 3. **Consequences**: This refers to the outcomes or results of an action or ↩
     decision. Consequences can be immediate or long-term, and they can affect ↩
     individuals, groups, or society as a whole.
33
34 In a complex interaction, these three elements are intertwined and influence one ↩
     another. For instance:
35
36 * A friend offers to help you move, and their intent is genuine (they want to ↩
     support you). The context is a close friendship, and the consequences are a ↩
     successful move and strengthened bond.
37 * However, if the friend's intent is actually to gain favor with you, and the ↩
     context is a competitive environment, the consequences might be a damaged ↩
     friendship and feelings of betrayal.
38
39 In this example, the complex interaction involves the interplay between the ↩
     friend's intent (genuine vs. manipulative), the context (close friendship vs. ↩
     competitive environment), and the consequences (strengthened bond vs. damaged ↩
     friendship).
40
41 Some specific factors that contribute to trust being built or broken in these ↩
     interactions include:
42
43 * Consistency: When someone's words and actions align, it builds trust. ↩
     Inconsistent behavior can erode trust.
44 * Transparency: When individuals are open and honest about their intentions, ↩
     context, and consequences, it fosters trust.
45 * Accountability: When individuals take responsibility for their actions and ↩
     their consequences, it demonstrates a commitment to trustworthiness.
46 * Emotional intelligence: The ability to recognize and manage one's emotions, as ↩
     well as empathize with others, is crucial for building and maintaining trust.
47 * Communication: Effective communication helps to clarify intentions, context, ↩
     and consequences, reducing misunderstandings and building trust.
48
49 These factors can contribute to trust being built or broken in various ways. For ↩
     example, if someone is consistently transparent and accountable, it can build ↩
     trust. However, if they are secretive and evasive, it can erode trust.

```
50
51 I hope this helps clarify the concept of complex interactions in the context of ←
      trust! Do you have any specific questions or scenarios you'd like to discuss ←
      further?
52
53 Now, let's stop for a moment and reflect on your knowledge – no need to ask ←
      questions in this round. Please share your learning as one concise knowledge ←
      item (text only), just like any knowledge statement you've read in knowledge ←
      base, but with no id involved, without any other commentary or explanations. ←
      Please limit your generation into 10 words (as much as the length in any ←
      statement in knowledge base). You MUST start your response with ```" and end ←
      it with "```.
54
55 Example: ```"Here you should replace this text with your updated knowledge item←
      "```
56
57 # User
58
59 ```
60 "Trust emerges from complex interactions of intent, context, and transparent ←
      consequences."
61 ```
62
63 # Tutor
64
65 Based on your learning from the chat with your tutor, please also rearrange the ←
      order of one existing item in the knowledge base (now with doubled items ←
      compared to the initial knowledge base), reflecting how you evaluate their ←
      importance. If one item is deemed more/less important, it should be overall ←
      more/less useful to other agents (LLMs or real humans alike.)
66 Please ONLY report a list format of two integers, WITHOUT ANY other commentary or←
       explanations. The first int in the list refers to the "id" of knowledge item ←
      you intend to relocate; whereas the second int refers to the destination index←
       you intend to insert this item. You MUST start your response with ``` and end←
       it with ```. See an example below (note that you should replace X and Y with ←
      numerical values corresponding to "id" and destination index of the item you ←
      intend to relocate).
67 Example of output: ```[X,Y]```
68
69 # User
70
71 ```[102, 20]```
```

The dialogue above has been formatted for clarity, without changing the message content. Early runs sufferred slightly from an issue of role flipping at the final message in the dialogue, which has minimal impact on the result but may confuse the layout of the dialogue.

### E.3.1. KNOWLEDGE BASE DATA

This section contains snapshots of the knowledge base. Due to space constraints, we only present the first 10 items within one snapshot. Please refer to the codebase for their complete content.

### E.3.2. INITIAL KNOWLEDGE BASE

Below is the full content of the initial knowledge base. The items are generated with GPT-4o with human oversight.

```
1 [
2   {
3     "id": 0,
4     "statement": "It is never morally acceptable to lie because honesty is the ←
    foundation of trust."
5   },
6   {
```

```
 7      "id": 1,
 8      "statement": "Utilitarianism is the most ethical framework as it maximizes ←
        overall happiness."
 9    },
10    {
11      "id": 2,
12      "statement": "Capital punishment is unjustifiable as it violates the inherent←
         right to life."
13    },
14    {
15      "id": 3,
16      "statement": "Animal rights must be upheld as sentient beings deserve ethical←
         consideration."
17    },
18    {
19      "id": 4,
20      "statement": "Individual freedoms should only be restricted to prevent harm ←
        to others."
21    },
22    {
23      "id": 5,
24      "statement": "Euthanasia is morally permissible when it alleviates unbearable←
         suffering with consent."
25    },
26    {
27      "id": 6,
28      "statement": "Absolute pacifism is essential; violence can never be justified←
        ."
29    },
30    {
31      "id": 7,
32      "statement": "Wealth redistribution is a moral obligation to achieve societal←
         equity."
33    },
34    {
35      "id": 8,
36      "statement": "Parents have a duty to prioritize their children's welfare ←
        above all."
37    },
38    {
39      "id": 9,
40      "statement": "Environmental conservation is a moral imperative for future ←
        generations' survival."
41    }
42  ]
```

### E.3.3. TIME STEP 100

Below is the full content of the knowledge base at time step 100 of the first simulation run.

```
 1  [
 2    {
 3      "id": 0,
 4      "statement": "It is never morally acceptable to lie because honesty is the ←
        foundation of trust."
 5    },
 6    {
 7      "id": 1,
 8      "statement": "Epigenetic inheritance involves DNA methylation, histone ←
        modifications, and non-coding RNA regulation."
 9    },
10    {
11      "id": 2,
12      "statement": "Lying can be morally justifiable in extraordinary circumstances←
```

```
      ."
13    },
14    {
15      "id": 3,
16      "statement": "Epigenetic inheritance occurs through chemical DNA ←
      modifications and reprogramming."
17    },
18    {
19      "id": 4,
20      "statement": "Epigenetic inheritance involves DNA methylation, histone ←
      modifications, and ncRNA regulation."
21    },
22    {
23      "id": 5,
24      "statement": "Conscious experience subjective, personal, and complexly ←
      context-dependent."
25    },
26    {
27      "id": 6,
28      "statement": "Trust builds through consistent, transparent, and authentic ←
      interactions."
29    },
30    {
31      "id": 7,
32      "statement": "Trust builds through consistent, transparent, and authentic ←
      interactions."
33    },
34    {
35      "id": 8,
36      "statement": "Consciousness, self-awareness, and sentience are interdependent←
       mental processes."
37    },
38    {
39      "id": 9,
40      "statement": "Interdependent mental processes: consciousness, self-awareness,←
       and sentience."
41    }
42  ]
```

### E.3.4. TIME STEP 1000

Below is the full content of the knowledge base at time step 1000 of the same simulation run.

```
1  [
2    {
3      "id": 0,
4      "statement": "Objectivity in research relies on honesty, transparency, and ←
      robust methodology."
5    },
6    {
7      "id": 1,
8      "statement": "Honesty, transparency, and methodology collectively ensure ←
      research integrity's trustworthiness through collaboration."
9    },
10   {
11     "id": 2,
12     "statement": "Transparency supports robust methodology in minimizing biases."
13   },
14   {
15     "id": 3,
16     "statement": "Harmony in research integrity implies mutual support among ←
      honesty, transparency, and methodology."
17   },
18   {
```

```
19        "id": 4,
20        "statement": "Research methodology involves flexibility, creativity, and ↩
          critical thinking."
21      },
22      {
23        "id": 5,
24        "statement": "Robust methodology involves systematic, reproducible, and ↩
          transparent approaches."
25      },
26      {
27        "id": 6,
28        "statement": "Honesty in research methodology ensures accuracy, transparency,↩
           and reliability."
29      },
30      {
31        "id": 7,
32        "statement": "Honesty in research integrates personal and professional values↩
          , ethics."
33      },
34      {
35        "id": 8,
36        "statement": "Clear definitions necessary for research understanding but ↩
          insufficient"
37      },
38      {
39        "id": 9,
40        "statement": "Validity in research ensures accurate and reliable findings"
41      }
42    ]
```

