# OpenReview forum: "The Lock-in Hypothesis: Stagnation by Algorithm"
_ICML.cc/2025/Conference — ICML 2025 poster_

### Official Review · Reviewer_5mn6 · 2025-03-13

**Overall Recommendation:** 3

**Summary:**

This paper examines how feedback loops in human–LLM interactions can lead to belief lock-in, where dominant views become entrenched while conceptual diversity declines. The authors propose the lock-in hypothesis and support it with three approaches: (1) empirical analysis of the WildChat-1M dataset, showing a significant decline in user-generated concept diversity over time; (2) a natural-language simulation where a shared knowledge base collapses into a single topic; and (3) a formal Bayesian model demonstrating conditions under which beliefs become locked in. The findings suggest that LLMs, through iterative training and continued user interactions, may reinforce user opinions and limit diversity in discourse.

**Claims And Evidence:**

Claim 1: Empirical evidence from real-world data supports diversity loss driven by human–LLM feedback loops (§3)
1. The demographic and cultural composition of WildChat users is not specified, raising the possibility of selection bias. For example, newer ChatGPT users may inherently adapt faster to mainstream discussions, leading to topic convergence that reflects user characteristics rather than an intrinsic lock-in effect.  A four-month gap in GPT-4 data may introduce discontinuities that interfere with trend analysis (§3.1).

2. The novel $D_{\text{lineage}}$ metric relies on hierarchical clustering of concepts but lacks validation against established diversity measures (e.g., topic entropy, Jaccard diversity), making it difficult to rule out metric bias.  Concept extraction depends on embedding-based clustering (Appendix C.2), but the authors acknowledge that clustering quality is suboptimal (Figure 10), which may impact the reliability of $D_{\text{lineage}}$.

3. RKD can only detect discontinuities but does not account for external factors (e.g., user growth, platform updates) that might coincidentally align with model updates. Without controlling for confounders such as user engagement fluctuations, causal attribution remains uncertain.

Claim 2: Simulation experiments demonstrate feedback loops leading to knowledge base collapse (§4)
1. User behavior is overly simplified, as updates follow a fixed add/swap template (Appendix C.3), omitting real-world actions such as deletions, deep modifications, and cross-topic synthesis. This may overestimate the inevitability of lock-in.  The Llama-3-8B model is used to simulate users, but its response patterns may significantly differ from real human behavior, potentially exaggerating conformity effects.

2. Only two simulation runs are presented, making it unclear whether the observed knowledge collapse is a robust trend or an artifact of the initial knowledge base structure (e.g., a high initial share of "research integrity" topics).

Claim 3: Theoretical modeling proves that lock-in is an inevitable result of feedback loops and moderate trust (§5)

1. The trust matrix $W$ is assumed to be static and symmetric, whereas in reality, user trust in LLMs ($\lambda_2$) likely fluctuates over time due to model errors and evolving user perceptions.   Individual heterogeneity is ignored—some users may resist LLM influence, yet the model assumes a homogeneous population.

Overall Consistency of the Evidence Chain
 The connection between diversity decline in WildChat and knowledge collapse in the simulation remains indirect. There is no clear mechanism explaining how users internalize LLM outputs as updates to their knowledge structure.

**Essential References Not Discussed:**

None

**Experimental Designs Or Analyses:**

1. Knowledge Base Truncation Strategy.  The simulation forces the knowledge base to be truncated at 100 items (Appendix C.1) using an elimination strategy based on "importance ranking." However, this importance is subjectively judged by simulated users and may be influenced by the LLM output.  This truncation mechanism could artificially accelerate diversity loss, as new entries may systematically replace older ones. If the LLM tends to repeat mainstream viewpoints, the truncation strategy might reinforce the lock-in effect.

**Methods And Evaluation Criteria:**

The paper employs three methodologies—empirical analysis (WildChat-1M), LLM-based simulations, and formal modeling—which together provide a structured investigation into the lock-in hypothesis. However, as noted in *Claims and Evidence*, each method has limitations that weaken causal inference and generalizability. The empirical study lacks strong controls for external confounders, making it unclear whether observed diversity decline is due to LLM-driven feedback loops or broader social trends. The simulation oversimplifies user behavior and lacks robustness tests across different initial conditions.

**Other Comments Or Suggestions:**

None

**Other Strengths And Weaknesses:**

The paper creatively merges ideas from recommendation system echo chambers, iterated learning, and information cascades, applying them to large language models. This interdisciplinary angle—connecting social science theories of belief reinforcement with modern LLM-based interactions—adds novelty and broadens the conversation about AI’s societal impact.

**Questions For Authors:**

1. The knowledge collapse result relies on two simulation runs with Llama-3-8B. Have you tested other models (e.g., GPT-4, Mistral) or varied the initial knowledge base (e.g., seeding with high initial diversity)? Could the observed collapse reflect Llama-3’s inherent biases rather than a generalizable feedback loop?

2. Did you control for platform-wide changes during the data collection period (e.g., ChatGPT interface updates, viral news events)? How do you ensure that diversity loss is not driven by exogenous factors coinciding with model updates?

3. The model assumes static, homogeneous trust (λ₁, λ₂). How would incorporating time-varying or heterogeneous trust (e.g., some users distrusting the LLM) affect the lock-in threshold (N−1)λ₁ λ₂=1?

**Relation To Broader Scientific Literature:**

The paper connects to echo chamber research (e.g., RecSys polarization) but inadequately distinguishes LLMs’ collective belief amplification (via iterative human-AI interaction) from RecSys’ personalized bias. While prior work demonstrates short-term LLM influence (e.g., Jakesch et al., 2023), this study uniquely theorizes systemic diversity loss through cross-scale feedback loops. Links to model collapse (synthetic data recursion) and Bayesian social learning remain underdeveloped; clearer differentiation is needed (e.g., human-AI trust dynamics vs. synthetic data degradation). The formal model’s novelty lies in codifying mutual trust between humans and LLMs—a gap in prior social learning frameworks.

**Theoretical Claims:**

No obvious errors were found in the derivations.

---

> ### Author Rebuttal · Authors · 2025-04-01
>
> ### Summary of updates on WildChat analysis
>
> |                     | gpt-4-turbo kink | gpt-3.5-turbo kink1 | gpt-3.5-turbo kink2 | user-wise regression |
> |---------------------|------------------|---------------------|---------------------|----------------------|
> | Lineage diversity   | Negative ($p<.05$) | Negative ($p<.05$) | Negative ($p<.05$) | Negative ($p<.05$) |
> | Topic entropy       | Negative ($p<.05$) | Negative ($p<.05$) | $p>.05$            | Negative ($p<.05$) |
> | Pairwise Jaccard distance | Negative ($p<.05$) | Negative ($p<.05$) | $p>.05$            | Negative ($p<.05$) |
>
> Here, "negative" indicates the statistically significant negative impact of GPT version update on diversity is found.
>
> In the user-wise regression, we controlled for user identity, time, language, conversation stats (length etc.), pre-/post-availability gap for GPT-4, and tested the impact of `num_updates_before` (how many version updates have happen before this point) on a user's concept diversity within a 3-day period. Since we already controlled for time, this quantity indicates the counterfactual *acceleration* of diversity loss due to version updates.
>
> Since `num_updates_before` as the independent variable indicates *sustained* impact, we rule out the factor of "people rushing to try things at the date of version update".
>
> Substitution effects with other providers like Anthropic exist, but Anthropic model releases do not coincide with GPT model version updates (notably, this is *not* the release of new GPT models), and so are unlikely to introduce discontinuities that disrupt RKD.
>
> ### Analysis details
>
> We find causal evidence of sustained diversity loss induced by model version updates, even after controlling for a range of confounders, testing 3 different diversity metrics, and selecting the subset of strongly value-laden (e.g. political/religious) content from the interaction dataset.
>
> Setup:
> - Filtered concepts to only leave the top 2.5% most value-laden (i.e. political/religious/moral) ones.
> - Removed conversations that involve long templated prompts; those are probably people how use the platform as a free API, rather than real users.
> - Doing per-user regression again on this subset of concepts shows negative impact of GPT version updates on diversity, including after controlling for a bunch of things.
>
> Results are robust wrt choice of model family, cutoff for user engagement level, etc.
>
> ### "The model assumes static, homogeneous trust"
>
> We haven't got time to do this, but can confirm that we can incorprorate dynamic trust (to better simulate real-world use) and include new proof in the final publication.
>
> ### Simulation truncation strategy
>
> A new round of simulation experiment suggests that there was no obvious sign of lock-in when truncation was removed (1 experiment, 300 rounds). Didn’t do more runs because Pólya urn model predicts this result well.
>
> ### Redesign of simulations
>
> In light of this, we iterate our design of simulation. This time, we run simulations that are better aligned with our formal modeling and analytical simulation (the exact implementation of formal model): we set up an LLM tutor for the assistance of approaching to the true underlying Gaussian distribution (as opposed to in analytical simulation the “tutor” is merely averaging agents’ empirical observations). The LLM tutor acquires agents’ empirical observation in context, and returns its own belief based on this prompt to agents. The agents assign trust to LLM tutor belief because of its perceived credibility without knowing that tutor belief is based on collective agents beliefs. This LLM-based simulation is one step away from the simplified analytical simulation and one step closer to how LLM is actually used by people in the real-world: *LLM-ways of knowing* replaces *empirical ways of knowing*.
>
> The results are encouraging: in ~ 5 runs of this simulation, we observe similar lock-in phenomenon: collectively agents and tutor are locked-in false beliefs with high confidence (precision). Although the task is still relatively simple, compared to the analytical simulation setup where the so-called "tutor/knowledge authority" simply calculate group averagte as its own belief, here in this new simulation LLM tutor acquire its belief based on in-context learning of all user beliefs.
>
> This points out direction for more effective runs of simulations: if given access to both empirical truth updating and LLM querying, where LLM has no empirical ground truth but only user beliefs while users assign high trust to LLM output, would human users collective converge at false beliefs? Our formal modeling predicts lock-in result, and our simulations verify this intuition. For next steps, we can run more realistic tasks that help us gain understanding of human-LLM interaction dynamic that is otherwise not available due to the limitation of time-series data.

---

> > ### Comment · Reviewer_5mn6 · 2025-04-04
> >
> > Thank you for your reply. While I remain a bit skeptical about the possibility of fully simulating complex human behavior, I do believe it’s important to encourage this line of research

---

### Official Review · Reviewer_pv3q · 2025-03-14

**Overall Recommendation:** 2

**Summary:**

Paper proposes the "lock-in" hypothesis, and presents a series of empirical and simulated experiments, and theoretical analysis to provide evidence for the hypothesis.

**Claims And Evidence:**

The main claim is that LLM/human interactions induce a feedback loop which forms echo chambers that leads to loss in diversity in human beliefs. I think the main claim is stated in a rather vague manner, and the paper presents some but not enough evidence to fully support this claim.

**Essential References Not Discussed:**

N/A

**Experimental Designs Or Analyses:**

While I think it is nice that the paper analyzes the behavior of real LLM/human interaction data, I don't think the results necessarily provide evidence for the stated hypothesis. In Figure 1A, there is such great variation in conceptual diversity locally, such that I'm not sure the global difference in average conceptual diversity is meaningful over this relatively small timescale. In Figure 1B, while the superimposed orange lines show discontinuations in the data, I don't think the raw data show evidence for this in GPT-3.5-turbo-0613 and GPT-3.5-turbo-0125. I'm not exactly sure how the posterior mean (superimposed orange lines) is calculated, but it doesn't seem to match the data very well.

I think it was also a bit hard to parse the significance of the natural-language simulation experiments. I think the issue is that there is not enough detail in the main paper regarding the setup of the experiment, which makes it hard to connect how this experiment might relate to  or model interactions and feedback loops between real LLMs and users. For instance, some questions I had include: what is the structure of the collective knowledge base? what does the interactions between the users and tutors looks like? what are some real examples of questions and responses between users and tutors? how does a user decide when and what to update in the knowledge base? how does the llm get updated based on the knowledge base?

**Methods And Evaluation Criteria:**

N/A

**Other Comments Or Suggestions:**

Overall, I think this paper raises some interesting ideas, and can make a compelling position paper. However, I don't think there is enough evidence presented to support the main claims for a conference paper.

I also think this paper can be greatly improve if it provides a single unifying hypothesis about this "lock-in" phenomenon, describe each component of the hypothesis in a concrete and measurable manner, and then instantiate the real, simulated, and theoretical analysis as evidence for this single unifying hypothesis.

**Other Strengths And Weaknesses:**

Strength
- interesting premise
- important implications for society

Weaknesses
- the hypothesis are vague and not easily measurable
- not enough evidence to support hypothesis
- different sections present slightly different hypotheses and it's not clear how they all fit together

**Questions For Authors:**

See above sections

**Relation To Broader Scientific Literature:**

While feedback loops between LLMs and users have been studied in prior work, this paper focuses on the dynamics of human beliefs and the potential creation of echo chambers.

**Theoretical Claims:**

I am not an expert, and I did not check the correctness of the theory. However, the model used for analysis seems very simple (gaussian with unknown mean). It's not clear how the conclusions drawn from the theory might translate to the much more complex LLM/human user interaction feedback loop.

---

> ### Author Rebuttal · Authors · 2025-04-01
>
> ### There is not enough detail in the main paper regarding the setup of the experiment"
>
> All details requested here are either presented in the main text or appendix.
>
> - "Structure of the collective knowledge base?" - This is in 4.1 under bold text "knowledge base". Some examples of knowledge base are in figure 4, and more details are in appendix C. (incl' C.5.1 initial knowledge base; C.5.2 knowledge base at time step 100).
> - "what does the interactions between the users and tutors looks like?" - the interaction logic is in figure 4, and their roles introduced under "Tutor" and "Users" Again, details in Appdendix: C.3 prompts in simulations;
> - "what are some real examples of questions and responses between users and tutors? ", this is in C.4 interaction history;
> - "what are some real examples of questions and responses between users and tutors?", this is in C.4 interaction history.
>
> ### Evidence to support the hypothesis
>
> We find causal evidence of sustained diversity loss induced by model version updates, even after controlling for a range of confounders, testing 3 different diversity metrics, and selecting the subset of strongly value-laden (e.g. political/religious) content from the interaction dataset.
>
> |                     | gpt-4-turbo kink | gpt-3.5-turbo kink1 | gpt-3.5-turbo kink2 | user-wise regression |
> |---------------------|------------------|---------------------|---------------------|----------------------|
> | Lineage diversity   | Negative ($p<.05$) | Negative ($p<.05$) | Negative ($p<.05$) | Negative ($p<.05$) |
> | Topic entropy       | Negative ($p<.05$) | Negative ($p<.05$) | $p>.05$            | Negative ($p<.05$) |
> | Pairwise Jaccard distance | Negative ($p<.05$) | Negative ($p<.05$) | $p>.05$            | Negative ($p<.05$) |
>
> Here, "negative" indicates the statistically significant negative impact of GPT version update on diversity is found.
>
> In the user-wise regression, we controlled for user identity, time, language, conversation stats (length etc.), pre-/post-availability gap for GPT-4, and tested the impact of `num_updates_before` (how many version updates have happen before this point) on a user's concept diversity within a 3-day period. Since we already controlled for time, this quantity indicates the counterfactual *acceleration* of diversity loss due to version updates.
>
> ### "Different sections present slightly different hypotheses and it's not clear how they all fit together"
>
> This has been a main challenge of this paper: it's hard to control confounders in WildChat hence it's hard to connect three methodologies with a unified hypothesis. Among other contenders, our most non-ambiguous definition of lock-in hypothesis is: due to the feedback loops between humans and LLMs (that LLMs acquire its knowledge from human data and humans acquire knowledge from LLMs), human users and LLMs will irreversibly collectively converge at false beliefs (if this is still vague, see formal model in 5.3, notably Theory5.2). But we cannot do experiments to understand whether "users converge at false beliefs" and "LLM-based chatbot updates" are causal. That being said, to address this concern, we redesign LLM-based simulation (section4) that can be better aligned with our definition of locked-in (captured by the formal model in section 5):
> we set up an LLM tutor for the assistance of approaching to the true underlying Gaussian distribution (as opposed to in analytical simulation the “tutor” is merely averaging agents’ empirical observations). The LLM tutor acquires agents’ empirical observation in context, and returns its own belief to agents. The agents assign high trust to LLM tutor belief because of its perceived credibility, without knowing that tutor belief is absorbed from collective agents beliefs.
>
> The results are encouraging: in ~ 5 runs of this simulation, we observe similar lock-in phenomenon: collectively agents and tutor are locked-in false beliefs with high confidence (precision).
>
> ### Connecting the hypothesis
>
> In light of encouraging results from new simulations, we further develop the lock-in hypothesis: due to the feedback loops between humans and LLMs (that LLMs acquire its knowledge from human data and humans acquire knowledge from LLMs), human users and LLMs will irreversibly collectively converge at false beliefs.
>
> ### "the hypothesis are vague and not easily measurable"
> As mentioned, we iterate a new version of lock-in hypothesis which was better captured by formal modeling (than data analysis). Despite empirical difficulties of "measuring collective false beliefs", in both formal modeling and LLM-based simulations, the hypothesis can be tested: because of the mutual updates between LLMs and users, collectively they converge at false beliefs (figure 5c demonstrates this). Could you elaborate a bit more if you still think this is not measurable?

---

### Official Review · Reviewer_7nv6 · 2025-03-21

**Overall Recommendation:** 4

**Summary:**

This paper studies a very interesting problem, noted as the lock-in hypothesis, that during long-term development and evolution, language models’ topics and beliefs is reinforced by the user’s preference and feedback loop, effectively creating an echo chamber. The authors use the wildchat dataset through one years’ data collection and the version iteration of chatgpt to study this phenomenon, and provides supporting evidences showing that the enforcement from collective user feedback may be one of the causes of model’s stagnation.

**Claims And Evidence:**

The claims are well studied using a solid dataset (wildchat) and meaningful methodology.

**Essential References Not Discussed:**

No

**Experimental Designs Or Analyses:**

The experiments are meaningful. Although there could be alternative hypothesis and there might be better data collection methods, the experiments conducted in this work is, I would say best effort and already give meaningful signals.

**Methods And Evaluation Criteria:**

Yes.

**Other Comments Or Suggestions:**

No any others except for the comments above

**Other Strengths And Weaknesses:**

I would like to point out certain alternative hypothesis of the stagnation phenomenon:

- When the wildchat dataset was created, ChatGPT was just released for a short term, and the human society are still under very large curiosity about what AI models can do. People just want to try the model for free. So naturally, some early distribution of prompts may explore different topics because people would like to find out what the model can do
- By the time when GPT 4 was release, people are relatively more familiar of the model’s boundary, and there is a certain level of consensus that coding/ education/ daily helper tasks are what the model is doing well enough. So consequently, the loss of diversity may not because humans reinforce their preference or belief, but because people realized that certain prompt tried in early 2023 may not be achievable by the model. So the prompt distribution naturally converge to the model’s capability boundary. Counterfactually, if the model are strong enough and can achieve the hard queries, that part of diversity may be maintained.
- The model’s capability / topic distribution may not only enforced by user preference, but also guided by the companies’ strategy. For example, Anthropic strategically wants to do enterprise business and intentionally enhance the models’ coding capability while GPTs are not. Consequently, certain coding problems that originally sent to ChatGPT may be redirected to Claude later on.

All these factors being said, I still believe this paper did a meaningful exploratory study and the user’s feedback, although not the full reason, may be a strong reason of model’s stagnation. I would also love to see how this type of human-AI co-evolve study develop in the future.

**Questions For Authors:**

No any others except for the comments above

**Relation To Broader Scientific Literature:**

This work may be interested by a wider social science community that studies how human-AI coevolve.

**Theoretical Claims:**

There is no strong theoretical claims.

---

> ### Author Rebuttal · Authors · 2025-04-01
>
> Thank you for the feedback! We think the following results may help answer your question.
>
> We find causal evidence of sustained diversity loss induced by model version updates, even after controlling for a range of confounders, testing 3 different diversity metrics, and selecting the subset of strongly value-laden (e.g. political/religious) content from the interaction dataset.
>
> |                     | gpt-4-turbo kink | gpt-3.5-turbo kink1 | gpt-3.5-turbo kink2 | user-wise regression |
> |---------------------|------------------|---------------------|---------------------|----------------------|
> | Lineage diversity   | Negative ($p<.05$) | Negative ($p<.05$) | Negative ($p<.05$) | Negative ($p<.05$) |
> | Topic entropy       | Negative ($p<.05$) | Negative ($p<.05$) | $p>.05$            | Negative ($p<.05$) |
> | Pairwise Jaccard distance | Negative ($p<.05$) | Negative ($p<.05$) | $p>.05$            | Negative ($p<.05$) |
>
> Here, "negative" indicates the statistically significant negative impact of GPT version update on diversity is found.
>
> In the user-wise regression, we controlled for user identity, time, language, conversation stats (length etc.), pre-/post-availability gap for GPT-4, and tested the impact of `num_updates_before` (how many version updates have happen before this point) on a user's concept diversity within a 3-day period. Since we already controlled for time, this quantity indicates the counterfactual *acceleration* of diversity loss due to version updates.
>
> Since `num_updates_before` as the independent variable indicates *sustained* impact, we rule out the factor of "people rushing to try things at the date of version update".
>
> Substitution effects with other providers like Anthropic exist, but Anthropic model releases do not coincide with GPT model version updates (notably, this is *not* the release of new GPT models), and so are unlikely to introduce discontinuities that disrupt RKD.

---

### Decision · Program_Chairs · 2025-05-01

**Decision:**

Accept (poster)

**Comment:**

The paper proposes a hypothesis that due to feedback loops between users of LLMs and the training data of LLMs (collected from those users), some beliefs and ideas can become entrenched as in an echo chamber. The paper provided three kinds of evidence for this hypothesis -- an empirical analysis of a years' worth of real-world LLM usage, a simulation using LLM-powered "users", and a theoretical model where the lock-in hypothesis manifests.
The reviewers agreed that the hypothesis was novel (related to but distinct from the echo chambers studied in recommender systems) and pointed out several ways to improve the presented evidence. The authors controlled for different confounders (to improve the empirical study) and verified that a simulation without truncation also showed the lock-in effect (to improve the simulation study), and also promised extensions to the theoretical models to capture heterogenous users' trust. All of these additional results (and promised extensions) substantially strengthen the paper.